# Multi-Objective Bandits with Hierarchical Preferences: A Thompson Sampling Approach

## Abstract

This paper studies multi-objective bandits with hierarchical preferences, a class of bandit problems where arms are evaluated according to multiple objectives, each with a distinct priority level. The agent aims to maximize the most critical objective first, followed by the second most important, and so on for subsequent objectives. We address this problem using Bayesian decision-making strategies. Although Bayesian methods have been extensively studied in single-objective bandit settings, its effectiveness in lexicographic bandits remains an open question. To fill this gap, we propose two TS-based algorithms for lexicographic bandits: **(i)** For Gaussian reward distributions, we introduce an multi-armed bandit algorithm that achieves a *problem-dependent regret bound* of $O(\sum_{\Delta^i(a) > 0} \frac{\log(mKT)}{\Delta^i(a)})$, where $\Delta^i(a)$ denotes the suboptimality gap for the objective $i \in [m]$ and arm $a \in [K]$, and $m$ is the number of objectives. **(ii)** For unknown reward distributions, we design a stochastic linear bandit algorithm with a *minmax regret bound* of $\widetilde{O}(d^{3/2}\sqrt{T})$, where $d$ is the dimension of the contextual vectors. These results highlight the adaptability of TS strategy to the lexicographic bandit problem, offering efficient solutions under varying degrees of knowledge about the rewards. Empirical experiments support our theoretical findings.

## 1 Introduction

Multi-armed bandits (MAB) is a sequential decision-making model under uncertainty, where an agent selects an arm (action) from an arm set $[K] = \{1, 2, \ldots, K\}$ and receives a stochastic reward corresponding to the chosen arm (Robbins, 1952; Lattimore & Szepesvári, 2020). The agent's goal is to maximize cumulative rewards, which requires balancing the exploration of uncertain arms with the exploitation of the best-known arms (Auer, 2002). This exploration-exploitation trade-off is central to a variety of applications, such as online advertising (Schwartz et al., 2017), recommendation systems (Li et al., 2010), and clinical trials (Villar et al., 2015), where decisions must be made under uncertainty. These scenarios often involve contextual information, which motivates the development of stochastic linear bandits (SLB) (Abbasi-yadkori et al., 2011; Chu et al., 2011; Jun & Kim, 2024). In SLB, arms are represented by feature vectors $\mathcal{A} \subseteq \mathbb{R}^d$, and the expected reward of each arm is a linear function of its features and an unknown parameter vector. In this paper, we first propose an algorithm for MAB with *finite* arms, and then present an algorithm for SLB with *infinite* arms.

In addressing the stochastic bandit problem, two widely studied and influential strategies are the Upper Confidence Bound (UCB) (Auer, 2002; Abbasi-yadkori et al., 2011) and Thompson Sampling (TS) (Agrawal & Goyal, 2013a; Russo & Van Roy, 2014; Xu et al., 2023; Clavier et al., 2024). UCB operates in the *frequentist* framework, assuming that the underlying parameters of the bandit model are fixed. At each round, it constructs confidence intervals for each arm's expected reward and selects the arm with the highest upper confidence bound (Abbasi-yadkori et al., 2011). In contrast, TS is based on the *Bayesian* framework, where the bandit parameters follow a prior distribution (Agrawal & Goyal, 2013b). At every step, TS samples a random value from the posterior distribution of each arm, and selects the arm with the highest sampled value. The *simplicity and near-optimal regret performance* of TS make it particularly appealing. Specifically, the UCB-based method (Abbasi-yadkori et al., 2011) requires solving a bilinear optimization problem at each round, which is non-convex and computationally demanding. By contrast, TS involves only linear optimization, significantly enhancing computational efficiency (Abeille & Lazaric, 2017). Empirical

Table 1: Comparisons of the Regret Bounds for TS Bandits.

| Algorithm | Regret | Obj. | Distribution | Model |
|---|---|---|---|---|
| Kaufmann et al. (2012b) | $O\left(\sum \frac{\log(T)}{\Delta(a)}\right)$ | Single | Known | MAB |
| Russo & Van Roy (2014) | $\widetilde{O}\left(d\sqrt{T}\right)$ | Single | Known | SLB |
| Agrawal & Goyal (2013a) | $\widetilde{O}\left(d^{3/2}\sqrt{T}\right)$ | Single | Unknown | SLB |
| Abeille & Lazaric (2017) | $\widetilde{O}\left(d^{3/2}\sqrt{T}\right)$ | Single | Unknown | SLB |
| DK-BULB (**Ours**) | $O\left((\Lambda^i(\lambda))^2 \cdot \sum \frac{\log(mKT)}{\Delta^i(a)}\right)$ | $i \in [m]$ | Known | MAB |
| DF-TSLB (**Ours**) | $\widetilde{O}\left(\Lambda^i(\lambda) \cdot d^{3/2}\sqrt{T}\right)$ | $i \in [m]$ | UnKnown | SLB |

studies further highlight the practical effectiveness of TS. While TS has been extensively studied in single-objective bandit problems (Agrawal & Goyal, 2013b), its application to multi-objective bandits remains relatively underexplored. However, many real-world scenarios necessitate the simultaneous optimization of multiple, potentially conflicting objectives. For instance, recommendation systems must balance user satisfaction (e.g., click or dwell time), platform revenue (e.g., purchase rate), and content diversity (Zheng & Wang, 2021). This highlights the importance of investigating the multi-objective bandit problem (Drugan & Nowe, 2013).

In multi-objective bandit problems, the rewards are vector-valued, which presents a challenge in comparing different arms. Existing approaches either utilize scalarization techniques to reduce the multi-objective problem to a single-objective one (Drugan & Nowe, 2013; Roijers et al., 2017; Wanigasekara et al., 2019), or apply Pareto dominance to identify multiple optimal arms (Auer et al., 2016; Lu et al., 2019; Xu & Klabjan, 2023; Crepon et al., 2024). However, scalarization methods *require precise knowledge* of the relative importance of objectives, while Pareto dominance *does not impose any priority ordering* across all objectives, which maybe violated in many real-world applications. For example, a hotel recommendation system prioritizes factors such as price, location, and service quality based on user preferences (Yager et al., 2011). Lexicographic bandits offer a framework that accommodates priority hierarchies, which first optimizes higher-priority objectives and then refines the lower-priority ones (Tekin & Turgay, 2018; Tekin, 2019; Hüyük & Tekin, 2021).

To the best of our knowledge, we are the first to design Bayesian algorithms for lexicographic bandit problems. Our main contributions are summarized as follows:

- For the MAB setting with Gaussian rewards, we propose an algorithm that achieves a regret bound of $O\left((\Lambda^i(\lambda))^2 \cdot \sum_{\Delta^i(a)>0} \frac{\log(mKT)}{\Delta^i(a)}\right)$ for the $i$-th objective, where $i \in [m]$, $\Lambda^i(\lambda) = 1 + \lambda + \cdots + \lambda^{i-1}$, $m$ is the number of objectives, $\lambda$ is the trade-off parameter among conflicting objectives, $\Delta^i(a)$ is the reward gap for arm $a$'s $i$-th objective, $K$ is the number of arms, and $T$ is the time horizon.

- For the SLB setting with unknown reward distributions, we propose an algorithm achieving a regret bound of $\widetilde{O}(\Lambda^i(\lambda) \cdot d^{3/2}\sqrt{T})$, where $d$ is the dimension of the contextual vector.

- As shown in Table 1, our algorithms yield regret bounds that are comparable to those of single-objective TS algorithms. Notably, since $\Lambda^1(\lambda) = 1$ for any $\lambda \in \mathbb{R}$, the performance of the most important objective is not degraded when optimizing the other objectives.

- We further provide an alternative proof for TS bandits, which differs from previous techniques that classify arms as saturated and unsaturated (Agrawal & Goyal, 2013a) or utilize the supporting functions (Abeille & Lazaric, 2017).

## 2 RELATED WORK

In this section, we provide a brief review of the literature on Thompson Sampling bandits and multi-objective bandits, highlighting key developments in both fields.

**Thompson Sampling Bandits.** Thompson Sampling (TS), first introduced by Thompson (1933), has become a fundamental approach for bandit problems, which is supported by extensive empirical (Scott, 2010; Chapelle & Li, 2011) and theoretical analysis (Kaufmann et al., 2012b; Agrawal & Goyal, 2013b; 2017). Existing TS algorithms can generally be classified into two categories: methods that assume known reward distributions (Kaufmann et al., 2012a;b; Russo & Van Roy, 2014; Atsidakou et al., 2023) and methods that are distribution-free (Agrawal & Goyal, 2012; 2013b;a; Abeille & Lazaric, 2017; Xu et al., 2023).

Kaufmann et al. (2012b) established an asymptotic regret bound of $O\left(\sum_{\Delta(a)>0} \frac{\log(T)}{\Delta(a)}\right)$ for MAB with Bernoulli rewards. This result was later extended by Kaufmann et al. (2012a) to various specific reward distributions. In a subsequent study, Russo & Van Roy (2014) proposed a regret bound of $\widetilde{O}(\sqrt{KT})$, assuming the posterior distribution is known. More recently, Atsidakou et al. (2023) derived a finite-time Bayes regret bound of $O\left(\sum_{\Delta(a)>0} \frac{\log(T)}{\Delta(a)}\right)$, applicable to both Gaussian and Bernoulli rewards. In cases where the reward distribution is unknown, Agrawal & Goyal (2012) demonstrated that the TS algorithm achieves an expected regret of $O\left((\sum_{\Delta(a)>0} \frac{1}{(\Delta(a))^2})^2 \log(T)\right)$ for the MAB model. Subsequently, Agrawal & Goyal (2013b) introduced a problem-independent regret bound of $\widetilde{O}(\sqrt{KT})$ for the MAB model. Agrawal & Goyal (2013a) proposed the first TS method for the SLB problem, proving a regret bound of $\widetilde{O}(d^{3/2}\sqrt{T})$ by categorizing arms into saturated and unsaturated groups. This result was refined by Abeille & Lazaric (2017), who revised the proof of Agrawal & Goyal (2013a) and obtained the same regret bound. Recently, Xu et al. (2023) developed a variance-aware TS algorithm for the SLB model.

**Multi-Objective Bandits.** Drugan & Nowe (2013) studied the single-objective MAB framework to multi-objective setting by associating a reward vector with each arm. Their work established logarithmic regret bounds under the scalarized regret and the Pareto regret, respectively, where the scalarized approach converts the multi-objective problem into a single-objective one by using weighted combinations of objectives, and the Pareto approach treats all objectives equally, without putting any weights on different objectives. Building on the Pareto approach, two lines of work are developed. One is Pareto regret minimization, which aims to minimize the cumulative Pareto regret over $T$ rounds (Turgay et al., 2018; Lu et al., 2019; Xu & Klabjan, 2023). Another research direction is the Pareto set identification, which aims to minimize the cost of identifying all Pareto optimal arms (Auer et al., 2016; Ararat & Tekin, 2023; Crepon et al., 2024). Most existing work on multi-objective Thompson sampling adopts the scalarized approach (Q. Yahyaa et al., 2015; Roijers et al., 2017; Paria et al., 2019), making it unsuitable for lexicographic bandit problems. Tekin & Turgay (2018) initially examined lexicographic contextual bandits with two objectives. Hüyük & Tekin (2021) extended the objectives beyond two in MAB model and achieved a priority-based regret bound of $\widetilde{O}((KT)^{2/3})$. Xue et al. (2024) studied the lexicographic Lipschitz bandit problem and proposed a regret bound of $\widetilde{O}(T^{(d_z^i+1)/(d_z^i+2)})$ for the $i$-th objective, where $d_z^i$ is the zooming dimension of the $i$-th objective and $i \in [m]$.

## 3 PROBLEM SETTING

This paper studies two multi-objective bandit model under lexicographic ordering: Multi-Objective Multi-Armed Bandits (MOMAB) and Multi-Objective Stochastic Linear Bandits (MOSLB).

**Notation.** For a vector $\boldsymbol{x} \in \mathbb{R}^d$, let $\|\boldsymbol{x}\|$ denote its Euclidean norm. Meanwhile, its norm induced by a positive-definite matrix $V \in \mathbb{R}^{d \times d}$ is $\|\boldsymbol{x}\|_V = \sqrt{\boldsymbol{x}^\top V \boldsymbol{x}}$. For any positive integer $m \in \mathbb{Z}_+$, $[m] \triangleq \{1, 2, \ldots, m\}$. The superscript $i \in [m]$ is used to distinguish different objectives, e.g., $y_t^i$ is the stochastic reward of the $i$-th objective at round $t$.

**MOMAB.** In the MOMAB problem, the arm set is $[K]$ and each arm $a \in [K]$ has a vector $[\mu^1(a), \mu^2(a), \ldots, \mu^m(a)] \in \mathbb{R}^m$. Here, $\mu^i(a)$ is the expected reward of arm $a$ for its $i$-th objective, and $m$ is the number of objectives. MOMAB is a $T$-round sequential decision-making problem. In each round $t = 1, 2, \ldots, T$, the agent chooses an arm $a_t \in [K]$ and receives a stochastic reward vector $[y_t^1, y_t^2, \ldots, y_t^m] \in \mathbb{R}^m$, where $\mathrm{E}[y_t^i] = \mu^i(a_t)$ for all $i \in [m]$. The lexicographic optimal arm is denoted as $a^*$ (we will define it later). For any arm $a \in [K]$ and $i \in [m]$, we set $\Delta^i(a) = \mu^i(a^*) - \mu^i(a)$. As in single-objective bandit problems (Lattimore & Szepesvári, 2020),

the agent's performance is measured by the cumulative reward gap over $T$ rounds, i.e.,

$$R^i(T) = \sum_{t=1}^{T} \Delta^i(a_t) = \sum_{t=1}^{T} \mu^i(a^*) - \mu^i(a_t), i \in [m].$$

**MOSLB.** In the MOSLB problem, the arm set at round $t$ is denoted as $\mathcal{A}_t \subseteq \mathbb{R}^d$, where $d$ is the dimension of contextual vector. In this paper, $\mathcal{A}_t$ is assumed to be infinite. There exist $m$ unknown vectors $\{\boldsymbol{\theta}_*^1, \boldsymbol{\theta}_*^2, \ldots, \boldsymbol{\theta}_*^m\} \subseteq \mathbb{R}^d$ which determine the expected rewards of each arm. Precisely, for each objective $i \in [m]$, the expected rewards for arm $\boldsymbol{x} \in \mathcal{A}_t$ is $\mu^i(\boldsymbol{x}) = \langle \boldsymbol{\theta}_*^i, \boldsymbol{x} \rangle$. It is often assumed that both the arms and inherent vectors are bounded, i.e.,

$$\|\boldsymbol{x}\| \le 1, \forall \boldsymbol{x} \in \mathcal{A}_t, \text{and } \|\boldsymbol{\theta}_*^i\| \le B, \forall i \in [m]. \tag{1}$$

In each round $t = 1, 2, \ldots, T$, the agent chooses an arm $\boldsymbol{x}_t \in \mathcal{A}_t$ and receives a stochastic reward vector associated with the chosen arm. Denote the lexicographic optimal arm in $\mathcal{A}_t$ as $\boldsymbol{x}_t^*$. The regret in MOSLB problem is written as

$$R^i(T) = \sum_{t=1}^{T} \langle \boldsymbol{\theta}_*^i, \boldsymbol{x}_t^* - \boldsymbol{x}_t \rangle, i \in [m].$$

Next, we introduce the lexicographic order to compare different arms (Hüyük & Tekin, 2021).

**Definition 1 (Lexicographic Order)** *Consider two vectors $\boldsymbol{u} = [u^1, u^2, \ldots, u^m] \in \mathbb{R}^m$ and $\boldsymbol{v} = [v^1, v^2, \ldots, v^m] \in \mathbb{R}^m$. $\boldsymbol{u}$ lexicographically dominates $\boldsymbol{v}$ if and only if there exists some $i^* \in [m]$ such that $u^i = v^i$ for $i \in [i^* - 1]$ and $u^{i^*} > v^{i^*}$.*

Lexicographic order compares vectors sequentially, starting with the first objective and proceeding to the last, e.g., $[3, 6, 2]$ lexicographically dominates $[3, 5, 10]$ and $i^* = 2$. Based on lexicographic order, we introduce the lexicographic optimal arm (Hüyük & Tekin, 2021).

**Definition 2 (Lexicographic Optimal Arm)** *An arm $a^* \in [K]$ or $\boldsymbol{x}_t^* \in \mathcal{A}_t$ is lexicographic optimal if and only if its expected reward is not lexicographically dominated by that of any other arms.*

To capture the trade-offs between conflicting objectives, we impose assumptions on the expected rewards. In the MOMAB setting, we assume that for any $i \ge 2$ and $a \in [K]$,

$$\mu^i(a) - \mu^i(a^*) \le \lambda \cdot \max_{j \in [i-1]} \{\mu^j(a_*) - \mu^j(a)\}. \tag{2}$$

A similar assumption for the MOSLB setting is that for any $i \ge 2$ and $\boldsymbol{x} \in \mathcal{A}_t$,

$$\langle \boldsymbol{\theta}_*^i, \boldsymbol{x} - \boldsymbol{x}_t^* \rangle \le \lambda \cdot \max_{j \in [i-1]} \langle \boldsymbol{\theta}_*^j, \boldsymbol{x}_t^* - \boldsymbol{x} \rangle, \quad i \in [m]. \tag{3}$$

Here, $\lambda$ quantifies the improvement in the value of the $i$-th objective for each unit decrease in the preceding $i - 1$ objectives, when the solution transitions from the optimal arm to other arms.

# 4 ALGORITHMS

In this section, we first present an algorithm for lexicographic MOMAB, and then introduce an algorithm for lexicographic MOSLB.

## 4.1 DISTRIBUTION-KNOWN METHOD: DK-BULB

This part provides a Distribution-Know BayesUCB method for Lexicographic Bandits, called DK-BULB, whose details are provided in Algorithm 1. We use Gaussian rewards for illustration in this paper, *and this method can be easily extended to other distributions*, such as Bernoulli rewards.

In the Gaussian MOMAB model, its inherent parameters $\{\theta_a^i | a \in [K], i \in [m]\} \subseteq \mathbb{R}$ are drawn from a known Gaussian prior distribution, which is

$$\theta_a^i \sim \mathcal{N}(\theta_{0,a}^i, \sigma_0^2), a \in [K], i \in [m], \tag{4}$$

---

**Algorithm 1** Distribution-Known BayesUCB for Lexicographic Bandits (DK-BULB)

---

**Input:** $T, K, m, \delta, \lambda, \{\theta_{0,a}^i | i \in [m], a \in [K]\}, \sigma_0, \sigma$

1: Initialize $\mathcal{A}_1 = [K]$
2: **for** $t = 1, 2, \ldots, T$ **do**
3:   Compute the posterior distribution $\mathcal{N}(\hat{\theta}_{t,a}^i, \hat{\sigma}_{t,a}^2)$ for any arm $a \in \mathcal{A}_t$ and objective $i \in [m]$,
     where $\hat{\theta}_{t,a}^i$ and $\hat{\sigma}_{t,a}^2$ are defined in Eq. (6)
4:   Compute the confidence term $c_t(a)$ for any arm $a \in \mathcal{A}_t$, where $c_t(a)$ is defined in Eq. (7)
5:   Choose the arm $a_t = \arg\max_{a \in \mathcal{A}_t} c_t(a)$
6:   Initialize the arm set $\mathcal{A}_t^0 = \mathcal{A}_t$
7:   **for** $i = 1, 2, \ldots, m$ **do**
8:     $\hat{a}_t^i = \arg\max_{a \in \mathcal{A}_t^{i-1}} \hat{\theta}_{t,a}^i$
9:     $\mathcal{A}_t^i = \{a \in \mathcal{A}_t^{i-1} | \hat{\theta}_{t,\hat{a}_t^i}^i - \hat{\theta}_{t,a}^i \leq (2 + 4\lambda + \cdots + 4\lambda^{i-1}) \cdot c_t(a_t)\}$
10:   **end for**
11:   Update $\mathcal{A}_{t+1} = \mathcal{A}_t^m$
12:   Play arm $a_t$ and observe its reward $[y_t^1, y_t^2, \ldots, y_t^m]$
13: **end for**

---

where $\theta_{0,a}^i \in \mathbb{R}$ is the prior mean and $\sigma_0 > 0$ is the prior standard deviation. For each arm $a \in [K]$ and each objective $i \in [m]$, its reward follows a Gaussian distribution:

$$y_a^i \sim \mathcal{N}(\theta_a^i, \sigma^2), \mu^i(a) = \theta_a^i, \tag{5}$$

where $\theta_a^i \in \mathbb{R}$ is the mean and $\sigma^2 > 0$ is the known variance.

DK-BULB adopts the idea of active arm elimination (AAE) to eliminate suboptimal arms during the $T$-round decision process. Unlike single-objective AAE algorithms (Even-Dar et al., 2006), DK-BULB has to deal with $m$ lexicographically prioritized objectives, which requires a hierarchical decision-making framework.

DK-BULB starts by initializing the candidate arm set $\mathcal{A}_1 = [K]$. In each round $t$, DK-BULB first uses historical data collected from previous rounds to compute the posterior distributions for current round. Leveraging a well-known result that the posterior distribution of a Gaussian random variable with a Gaussian prior is also Gaussian (Bishop, 2006), DK-BULB computes a Gaussian posterior distribution $\mathcal{N}(\hat{\theta}_{t,a}^i, \hat{\sigma}_{t,a}^2)$ for any arm $a \in \mathcal{A}_t$ and objective $i \in [m]$, where the posterior mean and posterior variance are defined as

$$\hat{\theta}_{t,a}^i = \hat{\sigma}_{t,a}^2 \left( \sigma_0^{-2} \theta_{0,a}^i + \sigma^{-2} \sum_{\tau=1}^{t-1} \mathbb{I}\{a_\tau = a\} y_t^i \right), \quad \hat{\sigma}_{t,a}^2 = \frac{1}{\sigma_0^{-2} + \sigma^{-2} N_{t,a}}. \tag{6}$$

Here, $N_{t,a} = \sum_{\tau=1}^{t-1} \mathbb{I}\{a_\tau = a\}$ denotes the number of observations for arm $a$ up to round $t$.

Based on the posterior variance, DK-BULB calculates the confidence term for arm $a$ as

$$c_t(a) = \sqrt{2\hat{\sigma}_{t,a}^2 \log(mKT/\delta)}, \tag{7}$$

which reflects the uncertainty in the posterior estimates. Next, the arm with maximal uncertainty among all eligible arms $\mathcal{A}_t$ is selected for further trials, i.e.,

$$a_t = \arg\max_{a \in \mathcal{A}_t} c_t(a). \tag{8}$$

To respect the lexicographic priority of the objectives, DK-BULB employs a hierarchical elimination mechanism. Beginning with the initial set of active arms $\mathcal{A}_t^0 = \mathcal{A}_t$, DK-BULB iteratively refines this set for each objective $i = 1, 2, \ldots, m$. At each refinement step, it identifies the arm $\hat{a}_t^i$ that maximizes the posterior mean $\hat{\theta}_{t,a}^i$ within the current active set $\mathcal{A}_t^{i-1}$, i.e., $\hat{a}_t^i = \arg\max_{a \in \mathcal{A}_t^{i-1}} \hat{\theta}_{t,a}^i$. Then, the active set is updated by retaining only those arms for which their posterior mean is sufficiently close to that of $\hat{a}_t^i$, such that

$$\mathcal{A}_t^i = \left\{ a \in \mathcal{A}_t^{i-1} | \hat{\theta}_{t,\hat{a}_t^i}^i - \hat{\theta}_{t,a}^i \leq (4\Lambda^i(\lambda) - 2) \cdot c_t(a_t) \right\},$$

where $\Lambda^i(\lambda) = 1 + \lambda + \cdots + \lambda^{i-1}$. Since $c_t(a_t)$ is the maximum confidence term among the currently active arms, the optimal arm $a^*$ remains in the active set.

After eliminating for all $m$ objectives, the active arm set for the next round is updated as $\mathcal{A}_{t+1} = \mathcal{A}_t^m$. Then, DK-BULB plays the arm $a_t$, and observes the corresponding rewards $[y_t^1, y_t^2, \ldots, y_t^m]$. These rewards are used to calculate the posterior mean and variance for subsequent rounds.

DK-BULB combines posterior estimation, confidence-based exploration, and lexicographic arm elimination to ensure that the selected arms adhere to the priority order of the objectives and balances exploration and exploitation. The upper bound on the regret of DK-BULB is provided as follows.

**Theorem 1** *Suppose that (2), (4) and (5) hold. Let $\Lambda^i(\lambda) = 1 + \lambda + \cdots + \lambda^{i-1}$. With probability at least $1 - \delta$, for any objective $i \in [m]$, the regret of DK-BULB satisfies*

$$R^i(T) \leq \sum_{\Delta^i(a)>0} \left( (4\Lambda^i(\lambda))^2 \sigma^2 \cdot \frac{2\log(mKT/\delta)}{\Delta_a^i} + \Delta_a^i \right).$$

**Remark 1** Theorem 1 states that for any objective $i \in [m]$, DK-BULB achieves a regret bound of $O\left( (\Lambda^i(\lambda))^2 \cdot \sum_{\Delta^i(a)>0} \frac{\log(mKT)}{\Delta^i(a)} \right)$, which is consistent with single-objective algorithms (Kaufmann et al., 2012b) in terms of $\Delta^i(a)$ and $T$. Although an additional term $\Lambda^i(\lambda)$ is included, this is the cost of optimizing multiple objectives simultaneously. $\Lambda^1(\lambda) = 1$ implies that when compared with single-objective algorithms (Kaufmann et al., 2012b), DK-BULB does not degrade the performance of the most important objective.

## 4.2 DISTRIBUTION-FREE METHOD: DF-TSLB

In this section, we introduce a Distribution-Free Thompson Sampling method for Lexicographic Bandits, referred to as DF-TSLB, with its details provided in Algorithm 2. DF-TSLB is specifically designed for the MOSLB model, and the only assumption on its rewards is that they satisfy the sub-Gaussian property. Specifically, for some $R > 0$ and any $\eta \in \mathbb{R}$, the following condition holds:

$$\mathrm{E}\left[ e^{\eta\left(y_t^i - \langle \boldsymbol{\theta}_*^i, \boldsymbol{x}_t \rangle\right)} | \boldsymbol{x}_t \right] \leq \exp\left( \frac{\eta^2 R^2}{2} \right), i \in [m]. \tag{9}$$

In the TS framework, the inherent parameters $\{\boldsymbol{\theta}_*^i\}_{i=1}^m$ are drawn from an unknown distribution, thus it is necessary to construct a posterior distribution based on historical data. Due to the linear structure of MOSLB, we estimate the mean of the posterior distribution by least squares estimation.

DF-TSLB begins by initializing the covariance matrix $V_1$ as the identity matrix $I \in \mathbb{R}^{d \times d}$ and sets the posterior mean for each objective to be zero vector, i.e, $\hat{\boldsymbol{\theta}}_1^i = \boldsymbol{0}, \forall i \in [m]$. At each round $t$, DF-TSLB first defines the confidence parameters $\alpha_t$ and $\beta_t$ to regulate exploration, as follows:

$$\alpha_t = R\sqrt{d\log(16mtT/\delta)} + B, \quad \beta_t = \alpha_t \cdot \sqrt{2d\log(8dmT/\delta)}. \tag{10}$$

Here, $\alpha_t$ quantifies the uncertainty in the least squares estimation and controls the variance of the posterior distribution. $\beta_t$ shows the uncertainty of the sampled estimators and guides exploration.

After setting the exploration parameters, for each objective $i \in [m]$, DF-TSLB samples an estimator $\tilde{\boldsymbol{\theta}}_t^i$ from a Gaussian distribution $\mathcal{N}(\hat{\boldsymbol{\theta}}_t^i, \alpha_t^2 \cdot V_t^{-1})$, where $\hat{\boldsymbol{\theta}}_t^i \in \mathbb{R}^d$ is the posterior mean derived from least squares estimation (Eq. (11)), and $V_t \in \mathbb{R}^{d \times d}$ is the covariance matrix. Using these sampled estimators, DF-TSLB engages in the decision-making process. It iteratively refines active arms, starting with $s = 1$ and the entire arm set at round $t$, $\mathcal{A}_{t,s} = \mathcal{A}_t$, until an arm is chosen.

Depending on the confidence term $\|\boldsymbol{x}\|_{V_t^{-1}}$ for candidate arms $\boldsymbol{x} \in \mathcal{A}_{t,s}$, the decision-making process is divided into three cases. **(i)** If $\|\boldsymbol{x}\|_{V_t^{-1}} \leq 1/\sqrt{T}$ for any $\boldsymbol{x} \in \mathcal{A}_{t,s}$, this indicates that all arms in $\mathcal{A}_{t,s}$ have been sufficiently explored. In this case, DF-TSLB first applies an arm elimination procedure, referred to as LAE, to filter out promising arms, and then randomly selects an arm $\boldsymbol{x}_t$ from the resulting set $\mathcal{A}_{t,T}$.

The detailed procedure of LAE is outlined in Algorithm 3. LAE eliminates arms using a procedure similar to Steps 6-11 in DK-BULB, which iteratively refines $\mathcal{A}_{t,s}$ for each objective $i \in [m]$. Starting

---

**Algorithm 2** Distribution-Free Thompson Sampling for Lexicographic Bandits (DF-TSLB)

---

**Input:** $T, d, m, \delta, \lambda, B$

1: Initialize $V_1 = I, \hat{\boldsymbol{\theta}}_1^i = \mathbf{0}$ for $i \in [m]$
2: **for** $t = 1, 2, \ldots, T$ **do**
3:   Set confidence parameters $\alpha_t$ and $\beta_t$ by Eq. (10)
4:   Sample $\tilde{\boldsymbol{\theta}}_t^i \sim \mathcal{N}(\hat{\boldsymbol{\theta}}_t^i, \alpha_t^2 \cdot V_t^{-1})$ for all $i \in [m]$
5:   Initialize $s = 1, \mathcal{A}_{t,s} = \mathcal{A}_t$
6:   **repeat**
7:     **if** $\|\boldsymbol{x}\|_{V_t^{-1}} \le 1/\sqrt{T}$ for any $\boldsymbol{x} \in \mathcal{A}_{t,s}$ **then**
8:       Run Algorithm 3 to obtain the promising arms: $\mathcal{A}_{t,T} = \text{LAE}(\tilde{\boldsymbol{\theta}}_t^i, \alpha_t, \beta_t, \mathcal{A}_{t,s}, 1/\sqrt{T})$
9:       Randomly choose an arm $\boldsymbol{x}_t \in \mathcal{A}_{t,T}$
10:     **else if** $\|\boldsymbol{x}_t\|_{V_t^{-1}} > 2^{-s}$ for some $\boldsymbol{x}_t \in \mathcal{A}_{t,s}$ **then**
11:       Choose the arm $\boldsymbol{x}_t$
12:     **else**
13:       Run Algorithm 3 to obtain the promising arms: $\mathcal{A}_{t,s+1} = \text{LAE}(\tilde{\boldsymbol{\theta}}_t^i, \alpha_t, \beta_t, \mathcal{A}_{t,s}, 2^{-s})$
14:       Update $s = s + 1$
15:     **end if**
16:   **until** an arm $\boldsymbol{x}_t$ is played
17:   Play arm $\boldsymbol{x}_t$ and observe its reward $[y_t^1, y_t^2, \ldots, y_t^m]$
18:   Update covariance matrix $V_{t+1} = V_t + \boldsymbol{x}_t \boldsymbol{x}_t^\top$
19:   Update $\hat{\boldsymbol{\theta}}_{t+1}^i = V_{t+1}^{-1} X_{t+1} Y_{t+1}^i$ for $i \in [m]$, where $X_{t+1}$ and $Y_{t+1}^i$ are defined in Eq. (12)
20: **end for**

---

**Algorithm 3** Lexicographic Arm Elimination (LAE)

---

**Input:** $\tilde{\boldsymbol{\theta}}_t^i, \alpha_t, \beta_t, \mathcal{A}_{t,s}, C$

1: Initialize the arm set $\mathcal{A}_{t,s}^0 = \mathcal{A}_{t,s}$
2: **for** $i = 1, 2, \ldots, m$ **do**
3:   $\hat{\boldsymbol{x}}_t^i = \arg\max_{\boldsymbol{x} \in \mathcal{A}_{t,s}^{i-1}} \langle \tilde{\boldsymbol{\theta}}_t^i, \boldsymbol{x} \rangle$
4:   $\mathcal{A}_{t,s}^i = \{\boldsymbol{x} \in \mathcal{A}_{t,s}^{i-1} | \langle \tilde{\boldsymbol{\theta}}_t^i, \hat{\boldsymbol{x}}_t^i - \boldsymbol{x} \rangle \le (2 + 4\lambda + \cdots + 4\lambda^{i-1}) \cdot (\alpha_t + \beta_t) \cdot C\}$
5: **end for**
6: Return $\mathcal{A}_{t,s}^m$

---

with the active arm set $\mathcal{A}_{t,s}^0 = \mathcal{A}_{t,s}$, LAE first identifies the arm that maximizes the posterior mean reward within the current active set $\mathcal{A}_{t,s}^{i-1}$, i.e., $\hat{\boldsymbol{x}}_t^i = \arg\max_{a \in \mathcal{A}_{t,s}^{i-1}} \langle \tilde{\boldsymbol{\theta}}_t^i, \boldsymbol{x} \rangle$. Then, the active set $\mathcal{A}_{t,s}^i$ retains only those arms whose difference between their posterior mean reward and that of $\hat{\boldsymbol{x}}_t^i$ does not exceed a threshold, i.e,

$$\langle \tilde{\boldsymbol{\theta}}_t^i, \hat{\boldsymbol{x}}_t^i - \boldsymbol{x} \rangle \le (2 + 4\lambda + \cdots + 4\lambda^{i-1}) \cdot (\alpha_t + \beta_t) \cdot C,$$

where $C$ is an exploration term that adapts as the decision-making process evolves. After eliminating for all $m$ objectives, LAE returns the active arm set $\mathcal{A}_t^m$.

**(ii)** If $\|\boldsymbol{x}_t\|_{V_t^{-1}} > 2^{-s}$ for some $\boldsymbol{x}_t \in \mathcal{A}_{t,s}$, the arm $\boldsymbol{x}_t$ is selected directly, as it has a high uncertainty and needs further exploration. **(iii)** If $\|\boldsymbol{x}\|_{V_t^{-1}} \le 2^{-s}$ for all $\boldsymbol{x} \in \mathcal{A}_{t,s}$, the set of promising arms $\mathcal{A}_{t,s}$ is refined using the LAE algorithm with the exploration term $C = 2^{-s}$. The index $s$ is then incremented ($s \to s + 1$), and the arm elimination process is repeated until an arm $\boldsymbol{x}_t$ is selected.

After the selected arm $\boldsymbol{x}_t$ is played and the corresponding rewards $[y_t^1, y_t^2, \ldots, y_t^m]$ are observed, DF-TSLB updates the posterior mean and variance to prepare for the decision of next round. Specifically, the covariance matrix is updated as $V_{t+1} = V_t + \boldsymbol{x}_t \boldsymbol{x}_t^\top$, and the posterior mean for each objective $i \in [m]$ is computed as

$$\hat{\boldsymbol{\theta}}_{t+1}^i = V_{t+1}^{-1} X_{t+1} Y_{t+1}^i, \tag{11}$$

where

$$X_{t+1} = [\boldsymbol{x}_1, \boldsymbol{x}_2, \ldots, \boldsymbol{x}_t], \quad Y_{t+1}^i = [y_1^i, y_2^i, \ldots, y_t^i]. \tag{12}$$

Finally, we present an upper regret bound for DF-TSLB.

**Theorem 2** *Suppose that (1), (3) and (9) hold. Let $\Lambda^i(\lambda) = 1 + \lambda + \cdots + \lambda^{i-1}$. With probability at least $1 - \delta$, for any objective $i \in [m]$, the regret of DF-TSLB satisfies*

$$R^i(T) \leq 44\Lambda^i(\lambda) \cdot (\alpha_T + \beta_T) \cdot \log(T) \cdot \sqrt{dT}.$$

**Remark 2** Theorem 2 states that DF-TSLB achieves a regret bound of $\widetilde{O}(\Lambda^i(\lambda) \cdot d^{3/2}\sqrt{T})$ for any objective $i \in [m]$. This result aligns with single-objective algorithms (Agrawal & Goyal, 2013a; Abeille & Lazaric, 2017) in terms of $d$ and $T$. The additional term $\Lambda^i(\lambda)$ captures the cost of optimizing multiple objectives simultaneously. $\Lambda^1(\lambda) = 1$ indicates that DF-TSLB does not degrade the performance of the most important objective. Additionally, it is noteworthy that our proof of Theorem 2 significantly differs from existing methods that classify arms as saturated or unsaturated (Agrawal & Goyal, 2013a) or utilize the properties of support functions (Abeille & Lazaric, 2017).

## 5 DK-BULB VS. DF-TSLB

Although DK-BULB and DF-TSLB share the common goal of addressing the lexicographic bandit problem, they differ significantly in the following aspects:

**Assumptions.** The primary distinction in the assumptions of our two algorithms is that DK-BULB requires distribution knowledge of the rewards and is designed for the MOMAB model, while DF-TSLB does not require such knowledge and is designed for the MOSLB model. Additionally, two other factors further differentiate these algorithms. First, DK-BULB assumes that the expected rewards $\{\theta_a^i | a \in [K], i \in [m]\}$ are drawn from a Gaussian prior distribution, making its expected rewards *unbounded*. In contrast, DF-TSLB satisfies the condition in Eq. (12), which ensures that its expected rewards are *bounded* by some constant $B > 0$. Second, DK-BULB assumes a *finite and fixed* arm set $\mathcal{A} = [K]$, whereas the arm set $\mathcal{A}_t$ of DF-TSLB can be *infinite and dynamic*.

**Implementation.** DK-BULB employs an *average sum* to estimate the posterior mean (Eq. (6)), whereas DF-TSLB utilizes *least squares estimation* (Eq. (11)). Besides, their strategies for arm selection also differ significantly in two key aspects. First, DK-BULB *selects the arm with the maximum confidence term* (Step 5), while DF-TSLB *divides the decision-making process into multiple stages, sequentially eliminating arms until a final choice is made* (Steps 5-16). This is due the arm set in DF-TSLB is changing, where new arms are continually added. Directly selecting the arm with the maximum confidence term in such a scenario would require excessive exploration, leading to increased regret. To address this, DF-TSLB alternates between exploration (Step 11) and exploitation (Step 13) across stages. Second, their *arm elimination thresholds* differ. For DK-BULB, the threshold is $(2 + 4\lambda + \cdots + \lambda^{i-1}) \cdot c_t(a_t)$, whereas for DF-TSLB, it is $(2 + 4\lambda + \cdots + \lambda^{i-1}) \cdot (\alpha_t + \beta_t) \cdot C$, and $C$ is dynamically adjusted during the decision-making process.

**Theorems.** Theorem 1 for DK-BULB provides a *problem-dependent* regret bound based on the expected reward gap, $\sum_{\Delta^i(a) > 0} \frac{1}{\Delta^i(a)}$, which adapts to specific problem instances. Specifically, a smaller positive gap $\Delta^i(a)$ indicates that the expected reward of a suboptimal arm $a$ is close to that of the optimal arm, making it more difficult to identify the optimal arm. However, this regret bound becomes invalid when $K$ is infinite. In contrast, Theorem 2 for DF-TSLB provides a regret bound with a different structure, emphasizing a growth rate of $d^{3/2}\sqrt{T}$. This bound is well-suited for the infinite-armed setting and captures the complexity of the high-dimensional context space.

Finally, we note that the $\lambda$-based hierarchical elimination mechanism originates from prior work (Xue et al., 2024). A central contribution of this paper is to show that Bayesian posterior-based exploration can be effectively integrated into this elimination framework to address lexicographic MOMAB and MOSLB, a direction not pursued in the earlier Lipschitz bandit setting (Xue et al., 2024). Our approach differs from existing studies (Xue et al., 2024; 2025) in three key aspects: (i) Algorithm 1 directly selects the arm with the largest posterior uncertainty (Step 5); (ii) it establishes instance-dependent regret guarantees that capture problem hardness; and (iii) it introduces a novel combination of BayesUCB/TS-style posterior exploration with $\lambda$-based hierarchical elimination, in contrast to their repeat-until search procedures, instance-independent bounds, and deterministic UCB-style analysis (Xue et al., 2024; 2025).

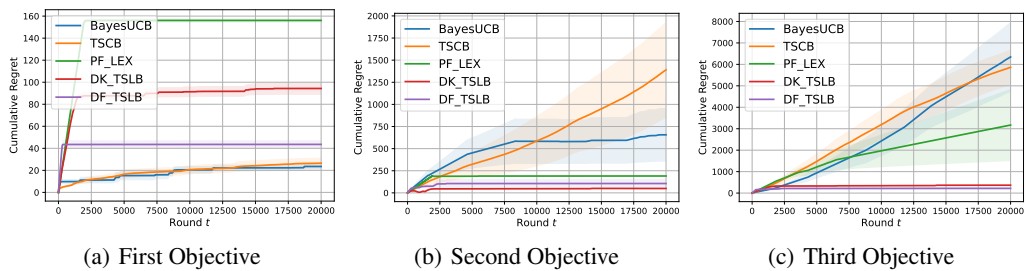

(a) First Objective       (b) Second Objective       (c) Third Objective

Figure 1: Comparison of our algorithms versus baselines. Each experiment is run 5 times, with average regret shown as lines and standard deviation as shaded areas.

## 6 EXPERIMENTS

This section presents empirical evaluations, where we compare our approaches against the lexicographic MOMAB algorithm PF-LEX (Hüyük & Tekin, 2021), as well as two single-objective algorithms: BayesUCB (Atsidakou et al., 2023) and TSCB (Agrawal & Goyal, 2013a). BayesUCB assumes knowledge of the reward distribution, whereas TSCB does not rely on this knowledge. Details of the experimental setup are in Appendix A.

Figure 1 shows the empirical performance of the baselines and our algorithms. Panels (a), (b), and (c) show the regret curves for the first, second, and third objectives, respectively. In Figure 1(a), BayesUCB and TSCB exhibit the lowest regret, as they are single-objective algorithms that focus solely on the first objective, thereby yielding optimal performance. Notably, the regret values of BayesUCB, TSCB, DK-BULB, and DF-TSLB are approximately 20, 20, 40, and 100, respectively. Given the long time horizon ($T = 20,000$), the regrets of DK-BULB and DF-TSLB remain only slightly higher than those of BayesUCB and TSCB.

Figure 1(b) presents the regret curves for the second objective, where DK-BULB and DF-TSLB clearly outperform the other methods. The regret curve for PF-LEX is higher than DK-BULB and DF-TSLB, which aligns with the theoretical guarantees. Specifically, the regret bound for PF-LEX is $\widetilde{O}((KT)^{2/3})$, whereas the regret bounds for DK-BULB and DF-TSLB are $O(K \log(KT))$ and $\widetilde{O}(d^{3/2}\sqrt{T})$, respectively. The regret curves for BayesUCB and TSCB continue to rise, indicating that these methods fail to identify the optimal arm and, consequently, cannot effectively optimize multiple objectives. Furthermore, the large deviations of BayesUCB and TSCB are attributed to the fact that these single-objective algorithms disregard the second objective, causing the second-objective rewards to appear random. Figure 1(c) shows the regret curves for the third objective. Once again, DK-BULB and DF-TSLB outperform all baseline methods, with their flat curves indicating successful identification of the lexicographic optimal arm.

## 7 CONCLUSION AND FUTURE WORK

This paper is the first to design Bayes-based algorithms for lexicographic bandits. **When** the rewards follow a Gaussian distribution, we propose an MOMAB algorithm that achieves a regret bound of $O\left((\Lambda^i(\lambda))^2 \cdot \sum_{\Delta^i(a)>0} \frac{\log(KT)}{\Delta^i(a)}\right)$ for any objective $i \in [m]$. Although our algorithm and analysis focuses on Gaussian rewards, Algorithm 1 can be easily extended to other distributions (e.g. Bernoulli rewards), as long as the posterior distribution is computable. **When** the reward distributions are unknown, we propose an MOSLB algorithm that achieves a regret bound of $\widetilde{O}(\Lambda^i(\lambda) \cdot d^{3/2}\sqrt{T})$ for any objective $i \in [m]$. Meanwhile, we provide an alternative proof for linear TS bandits, which differs from previous techniques that classify arms as saturated and unsaturated (Agrawal & Goyal, 2013a) or utilize the properties of support functions (Abeille & Lazaric, 2017).

Although our methods achieve comparable regret bounds to single-objective algorithms (Kaufmann et al., 2012b; Abeille & Lazaric, 2017) in term of $\Delta^i(a)$, $K$, $d$ and $T$, a challenging open problem is to remove the additional term $\Lambda^i(\lambda)$.

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

# A    EXPERIMENTAL SETTINGS

**MOMAB.** In the MOMAB setting, we set number of arms $K = 10$ and the number of objectives $m = 3$. For any arm $a \in [K]$, its expected rewards are defined as $\mu^1(a) = 1 - \min_{p \in \{0.3, 0.6, 0.9\}} |0.1 \times a - p|$, $\mu^2(a) = 1 - 2 \times \min_{p \in \{0.5, 0.8\}} |0.1 \times a - p|$, $\mu^3(a) = 1 - 2 \times |0.1 \times a - 0.5|$. The optimal arms for the first objective are $\{3, 6, 9\}$, and the optimal arms for both the first and second objectives are $\{6, 9\}$. Thus, to identify the lexicographic optimal arm $a^* = 6$, it is necessary to consider all three objectives.

**MOSLB.** In the MOSLB setting, we fix the arm set as $\mathcal{A}_t = \{\tilde{\boldsymbol{x}}_1, \tilde{\boldsymbol{x}}_2, \ldots, \tilde{\boldsymbol{x}}_K\} \subseteq \mathbb{R}^d$ for any $t \geq 1$. Both the arm number $K$ and feature dimension $d$ are set as 10, which ensures that MOMAB and MOSLB encounter the same number of unknown parameters. For $k \in [K]$, the arm vector $\tilde{\boldsymbol{x}}_k$ is set as the standard basis in $\mathbb{R}^d$, whose $k$-th element is 1 and all other elements are 0. The number of objectives is set as $m = 3$. We denote the inherent vectors as $\boldsymbol{\theta}_*^i = [\theta_*^i(1), \theta_*^i(2), \ldots, \theta_*^i(10)], i \in [3]$. The elements of $\boldsymbol{\theta}_*^1$, $\boldsymbol{\theta}_*^2$ and $\boldsymbol{\theta}_*^3$ are specified as $\theta_*^1(k) = 1 - \min_{p \in \{0.3, 0.6, 0.9\}} |0.1 \times k - p|$, $\theta_*^2(k) = 1 - 2 \times \min_{p \in \{0.5, 0.8\}} |0.1 \times k - p|$ and $\theta_*^3(k) = 1 - 2 \times |0.1 \times k - 0.5|, k \in [10]$. Thus, its expected rewards are the same as the MOMAB setting, enabling a direct comparison between the two models.

Although Algorithm 2 is capable of handling infinite arm sets, we use a finite arm set in the MOSLB experiments for the following reasons:

1. Using a finite arm set allows both MOMAB and MOSLB to be evaluated on the same problem instance, making the empirical comparison more meaningful and controlled.

2. Even if the arm set were infinite, in practice we would still construct a structured arm set (e.g., a ball or grid) so that the maximization steps in the algorithms admit exact solutions. This setup is conceptually equivalent to working with a finite discrete arm set.

3. Many existing stochastic linear bandit works conduct experiments on finite arm sets for the same practical reasons (Kim et al., 2021; Xu et al., 2023). Hence, our setup follows standard empirical practice in this domain.

All experiments were conducted on a Windows 10 laptop with an Intel(R) Core(TM) i7-1170 CPU and 32GB of RAM. Each algorithm was run with $\delta = 0.01$ and $T = 20,000$. The stochastic rewards $\{y_t^i\}_{t \in [T]}$ are drawn from a normal distribution with mean $\mu^i(a)$ or $\mu^i(\boldsymbol{x})$ and variance 0.1. Following the existing bandit work (Chapelle & Li, 2011; Jun et al., 2017), we scale the confidence terms for all algorithms by a factor selected from the range $[0.01, 1]$.

# B    PROOF OF THEOREM 1

Recall from Eq. (5) that, in the Gaussian MOMAB model, the expected reward for any arm $a \in [K]$ and any objective $i \in [m]$ is $\mu^i(a) = \theta_a^i$. Therefore, the regret for MOMAB can be rewritten as

$$R^i(T) = \sum_{t=1}^{T} \Delta^i(a_t) = \sum_{t=1}^{T} \theta_{a^*}^i - \theta_{a_t}^i, i \in [m].$$

Let $\mathcal{E}$ be the event

$$\mathcal{E} = \left\{ \forall t \in [T], \forall a \in [K], \forall i \in [m] : |\theta_a^i - \hat{\theta}_{t,a}^i| \leq c_t(a) \right\}, \tag{13}$$

where $\hat{\theta}_{t,a}^i$ is the posterior mean as calculated in Eq. (6), and $c_t(a)$ is the confidence term defined in Eq. (7).

To establish a foundation for the proof, we first introduce a lemma to show that the event $\mathcal{E}$ holds with high probability.

**Lemma 1 (Abramowitz (1964))** *For a Gaussian distributed random variable $Z$ with mean $m$ and variance $\sigma^2$, for any $z \geq 1$,*

$$\Pr\{|Z - m| > z\sigma\} \leq \frac{1}{\sqrt{\pi} z} e^{-z^2/2}.$$

Given that $\theta_a^i \sim \mathcal{N}(\hat{\theta}_{t,a}^i, \hat{\sigma}_{t,a}^2)$ and $c_t(a) = \sqrt{2\hat{\sigma}_{t,a}^2 \log(mKT/\delta)}$, we have for a fixed $t \in [T], a \in [K]$ and $i \in [m]$,

$$|\theta_a^i - \hat{\theta}_{t,a}^i| \le c_t(a)$$

holds with probability at least $1 - \frac{\delta}{mKT}$. Taking the union over all $t \in [T], a \in [K]$ and $i \in [m]$, we conclude that event $\mathcal{E}$ holds with probability at least $1 - \delta$.

The prioritized elimination mechanism in Steps 6 to 10 of Algorithm 1 is crucial for selecting arms in accordance with the priority order of the objectives, while efficiently balancing exploration and exploitation. We now present the following lemma, which demonstrates that the elimination mechanism in Algorithm 1 does not discard the lexicographic optimal arm, and that the remaining arms are promising.

**Lemma 2** Suppose $\mathcal{E}$ in Eq. (13) holds. In Steps 6 to 10 of Algorithm 1, if $a^* \in \mathcal{A}_t^0$, then

$$a_* \in \mathcal{A}_t^m \text{ and } \Delta^i(a) \le 4(1 + \lambda + \cdots + \lambda^{i-1}) \cdot c_t(a_t), \forall i \in [m], \forall a \in \mathcal{A}_t^m.$$

**Proof:** Given that the arm is eliminated from the 1-st objective to the $m$-th objective, we prove this lemma using an inductive approach. For the first objective $i = 1$, since $\hat{a}_t^1 = \arg\max_{a \in \mathcal{A}_t^0} \hat{\theta}_{t,a}^1$ and $a^* \in \mathcal{A}_t^0$, it follows that for all $a \in \mathcal{A}_t^1$,

$$\Delta^1(a) = \theta_{a^*}^1 - \theta_a^1 \le \theta_{a^*}^1 - \hat{\theta}_{t,a^*}^1 + \hat{\theta}_{t,\hat{a}_t^1}^1 - \theta_a^1. \tag{14}$$

Given that the event $\mathcal{E}$ holds, we have for all $a \in \mathcal{A}_t^1$,

$$\theta_{a^*}^1 - \hat{\theta}_{t,a^*}^1 \le c_t(a^*), \quad \hat{\theta}_{t,a}^1 - \theta_a^1 \le c_t(a).$$

Substituting these bounds into Eq. (14), we obtain for all $a \in \mathcal{A}_t^1$,

$$\Delta^1(a) \le c_t(a^*) + \hat{\theta}_{t,\hat{a}_t^1}^1 - \hat{\theta}_{t,a}^1 + c_t(a).$$

Recalling that $\mathcal{A}_t^1 = \{a \in \mathcal{A}_t^0 | \hat{\theta}_{t,\hat{a}_t^1}^1 - \hat{\theta}_{t,a}^1 \le 2c_t(a_t)\}$, it follows that for all $a \in \mathcal{A}_t^1$,

$$\Delta^1(a) \le c_t(a^*) + 2c_t(a_t) + c_t(a). \tag{15}$$

Since $a_t = \arg\max_{a \in \mathcal{A}_t^0} c_t(a)$, we have $c_t(a) \le c_t(a_t)$ for all $a \in \mathcal{A}_t^1 \subseteq \mathcal{A}_t^0$. Substituting this into Eq. (15) yields,

$$\Delta^1(a) \le 4c_t(a_t), \forall a \in \mathcal{A}_t^1.$$

Next, since the event $\mathcal{E}_t$ holds, we have

$$\hat{\theta}_{t,\hat{a}_t^1}^1 - \hat{\theta}_{t,a^*}^1 \le \theta_{\hat{a}_t^1}^1 + c_t(\hat{a}_t^1) - \theta_{a^*}^1 + c_t(a^*).$$

Given that $a^*$ is the optimal arm, it follows that $\theta_{\hat{a}_t^1}^1 - \theta_{a^*}^1 \le 0$. Reusing $c_t(a) \le c_t(a_t)$ for all $a \in \mathcal{A}_t^1 \subseteq \mathcal{A}_t^0$, we conclude

$$\hat{\theta}_{t,\hat{a}_t^1}^1 - \hat{\theta}_{t,a^*}^1 \le \theta_{\hat{a}_t^1}^1 + c_t(\hat{a}_t^1) - \theta_{a^*}^1 + c_t(a^*) \le 2c_t(a_t).$$

Thus, $a^* \in \mathcal{A}_t^1 = \{a \in \mathcal{A}_t^0 | \hat{\theta}_{t,\hat{a}_t^1}^1 - \hat{\theta}_{t,a}^1 \le 2c_t(a_t)\}$, completing the proof for the first objective.

By induction, for $i \ge 2$, assume

$$a^* \in \mathcal{A}_t^j \text{ and } \Delta^j(a) \le 4(1 + \lambda + \cdots + \lambda^{j-1}) \cdot c_t(a_t), \forall a \in \mathcal{A}_t^j, \forall j \in [i-1].$$

We aim to prove

$$a^* \in \mathcal{A}_t^i \text{ and } \Delta_a^i \le 4(1 + \lambda + \cdots + \lambda^{i-1}) \cdot c_t(a_t), \forall a \in \mathcal{A}_t^i. \tag{16}$$

Since $\hat{a}_t^i = \arg\max_{a \in \mathcal{A}_t^{i-1}} \hat{\theta}_{t,a}^i$ and $a^* \in \mathcal{A}_t^{i-1}$, it follows that for all $a \in \mathcal{A}_t^i \subseteq \mathcal{A}_t^{i-1}$,

$$\Delta^i(a) = \theta_{a^*}^i - \theta_a^i \le \theta_{a^*}^i - \hat{\theta}_{t,a^*}^i + \hat{\theta}_{t,\hat{a}_t^i}^i - \theta_a^i. \tag{17}$$

Given that event $\mathcal{E}$ holds, we have that for all $a \in \mathcal{A}_t^i$,

$$\theta_{a^*}^i - \hat{\theta}_{t,a^*}^i \leq c_t(a^*), \quad \hat{\theta}_{t,a}^i - \theta_a^i \leq c_t(a).$$

Substituting these bounds into Eq. (17) gives, for all $a \in \mathcal{A}_t^i$,

$$\Delta^i(a) \leq c_t(a^*) + \hat{\theta}_{t,\hat{a}^i}^i - \hat{\theta}_{t,a}^i + c_t(a).$$

Recalling that $\mathcal{A}_t^i = \{a \in \mathcal{A}_t^0 | \hat{\theta}_{t,\hat{a}_t^i}^i - \hat{\theta}_{t,a}^i \leq (2 + 4\lambda + \cdots + 4\lambda^{i-1}) \cdot c_t(a_t)\}$, it follows that for all $a \in \mathcal{A}_t^i$,

$$\Delta^i(a) \leq c_t(a^*) + (2 + 4\lambda + \cdots + 4\lambda^{i-1}) \cdot c_t(a_t) + c_t(a). \tag{18}$$

Since $a_t = \arg\max_{a \in \mathcal{A}_t^0} c_t(a)$, we have $c_t(a) \leq c_t(a_t)$ for all $a \in \mathcal{A}_t^i \subseteq \mathcal{A}_t^0$. Substituting this into Eq. (18) yields, for all $a \in \mathcal{A}_t^i$,

$$\Delta^i(a) \leq 4(1 + \lambda + \cdots + \lambda^{i-1}) \cdot c_t(a_t).$$

Next, since the event $\mathcal{E}_t$ holds, we have

$$\hat{\theta}_{t,\hat{a}_t^i}^i - \hat{\theta}_{t,a^*}^i \leq \theta_{\hat{a}_t^i}^i + c_t(\hat{a}_t^i) - \theta_{a^*}^i + c_t(a^*). \tag{19}$$

According to Eq. (2), $\theta_{\hat{a}_t^i}^i - \theta_{a^*}^i \leq \lambda \cdot \max_{j \in [i-1]}\{\theta_{a^*}^j - \theta_{\hat{a}_t^i}^j\}$. Thus,

$$\theta_{\hat{a}_t^i}^i - \theta_{a^*}^i \leq \lambda \cdot 4(1 + \lambda + \cdots + \lambda^{i-2}) \cdot c_t(a_t).$$

Reusing $c_t(a) \leq c_t(a_t)$ for all $a \in \mathcal{A}_t^i \subseteq \mathcal{A}_t^0$, taking this into Eq. (19) gives

$$\hat{\theta}_{t,\hat{a}_t^i}^i - \hat{\theta}_{t,a^*}^i \leq 4(\lambda + \lambda^2 + \cdots + \lambda^{i-1}) \cdot c_t(a_t) + 2c_t(a_t).$$

Thus, $a^* \in \mathcal{A}_t^i = \{a \in \mathcal{A}_t^{i-1} | \hat{\theta}_{t,\hat{a}_t^i}^i - \hat{\theta}_{t,a^*}^i \leq (2 + 4\lambda + \cdots + 4\lambda^{i-1}) \cdot c_t(a_t)\}$. Hence, Eq. (16) is proved, completing the induction framework and the proof of Lemma 2. $\qquad\square$

Lemma 2 depends on the assumption $a_* \in \mathcal{A}_t^0$. In the following lemma, we remove this assumption.

**Lemma 3** *Suppose $\mathcal{E}$ in Eq. (13) holds. In Algorithm 1, for any $a \in \mathcal{A}_{t+1}$,*

$$\Delta^i(a) \leq 4(1 + \lambda + \cdots + \lambda^{i-1}) \cdot c_t(a_t).$$

**Proof:** We prove by induction that $a^* \in \mathcal{A}_t^0$ for $t \geq 1$. For the base case $t = 1$, $a^* \in \mathcal{A}_1^0$ obviously since $\mathcal{A}_1^0 = [K]$. Now, assume $a^* \in \mathcal{A}_t^0$ for some $t \geq 1$. By Lemma 2, $a^* \in \mathcal{A}_t^0$ deduces that $a^* \in \mathcal{A}_t^m$. Given that $\mathcal{A}_{t+1}^0 = \mathcal{A}_{t+1} = \mathcal{A}_t^m$, it follows that $a^* \in \mathcal{A}_{t+1}^0$. Thus, by induction $a_* \in \mathcal{A}_t^0$ holds for all $t \geq 1$. With $a^* \in \mathcal{A}_t^0$, Lemma 2 tells that for any $a \in \mathcal{A}_t^m$, $\Delta^i(a) \leq 4(1 + \lambda + \cdots + \lambda^{i-1}) \cdot c_t(a_t)$ for $i \in [m]$. Therefore, Lemma 3 holds as $\mathcal{A}_t^m = \mathcal{A}_{t+1}$. $\square$

We now proceed to complete the proof of Theorem 1. For clarity, define $\Lambda^i(\lambda) = 1 + \lambda + \cdots + \lambda^{i-1}$ for any $i \in [m]$. Since $a_t \in \mathcal{A}_t$, Lemma 3 tells that

$$\Delta^i(a_t) \leq 4\Lambda^i(\lambda) \cdot c_{t-1}(a_{t-1}) = 4\Lambda^i(\lambda) \cdot \sqrt{\frac{2\log(mKT/\delta)}{\sigma_0^{-2} + \sigma^{-2}N_{t-1,a_{t-1}}}}. \tag{20}$$

From Step 5 of Algorithm 1, where $a_t = \arg\max_{a \in \mathcal{A}_t} c_t(a) = \arg\min_{a \in \mathcal{A}_t} N_{t,a}$, we have

$$N_{t,a_t} \leq N_{t,a_{t-1}} = N_{t-1,a_{t-1}} + 1.$$

Substituting this into Eq. (20), we have

$$\Delta^i(a_t) \leq 4\Lambda^i(\lambda) \cdot \sqrt{\frac{2\log(KT/\delta)}{\sigma_0^{-2} + \sigma^{-2}(N_{t,a_t} - 1)}}.$$

Reorganizing the inequality yields

$$N_{t,a_t} \leq (4\Lambda^i(\lambda))^2\sigma^2 \cdot \frac{2\log(mKT/\delta)}{(\Delta^i(a_t))^2} - \sigma^2\sigma_0^{-2} + 1 \leq (4\Lambda^i(\lambda))^2\sigma^2 \cdot \frac{2\log(mKT/\delta)}{(\Delta^i(a_t))^2} + 1.$$

Using the number of times each arm is played, we can bound the regret as follows:

$$R^i(T) = \sum_{t=1}^{T} \Delta^i(a_t) \le \sum_{\Delta^i(a)>0} \Delta^i(a) \cdot N_{T+1,a}$$

$$\le \sum_{\Delta^i(a)>0} \left( (4\Lambda^i(\lambda))^2 \sigma^2 \cdot \frac{2\log(mKT/\delta)}{\Delta_a^i} + \Delta_a^i \right).$$

Since the event $\mathcal{E}$ holds with probability at least $1 - \delta$, the above regret bound holds with the same probability. The proof of Theorem 1 is finished. $\square$

## C    PROOF OF THEOREM 2

We begin by presenting a lemma that establishes the confidence parameters in Eq. (10).

**Lemma 4** *With probability at least $1 - \delta$, for any $i \in [m]$ and $t \in [T]$,*

$$\|\hat{\boldsymbol{\theta}}_t^i - \boldsymbol{\theta}_*^i\|_{V_t} \le \alpha_t = R\sqrt{d\log\left(\frac{16mtT}{\delta}\right)} + B, \quad \|\tilde{\boldsymbol{\theta}}_t^i - \hat{\boldsymbol{\theta}}_t^i\|_{V_t} \le \beta_t = \alpha_t \cdot \sqrt{2d\log\left(\frac{8dmT}{\delta}\right)}.$$

**Proof:** For a fixed objective $i \in [m]$, Lemma 1 of Abeille & Lazaric (2017) guarantees that, with probability at least $1 - \delta$, for any round $t \in [T]$,

$$\|\hat{\boldsymbol{\theta}}_t^i - \boldsymbol{\theta}_*^i\|_{V_t} \le \tilde{\alpha}_t = R\sqrt{d\log\left(\frac{16tT}{\delta}\right)} + B, \quad \|\tilde{\boldsymbol{\theta}}_t^i - \hat{\boldsymbol{\theta}}_t^i\|_{V_t} \le \tilde{\beta}_t = \tilde{\alpha}_t \cdot \sqrt{2d\log\left(\frac{8dT}{\delta}\right)}.$$

Applying a union bound over all $i \in [m]$ finishes the proof of Lemma 4. $\square$

Define the event

$$\tilde{\mathcal{E}} = \left\{\forall t \in [T], \forall i \in [m] : \|\hat{\boldsymbol{\theta}}_t^i - \boldsymbol{\theta}_*^i\|_{V_t} \le \alpha_t, \|\tilde{\boldsymbol{\theta}}_t^i - \hat{\boldsymbol{\theta}}_t^i\|_{V_t} \le \beta_t\right\}, \tag{21}$$

From Lemma 4, it follows that event $\tilde{\mathcal{E}}$ holds with probability at least $1 - \delta$. Using this result, the posterior rewards can be bounded as follows.

**Lemma 5** *Suppose event $\tilde{\mathcal{E}}$ in Eq. (21) holds. For any $i \in [m]$ and $t \in [T]$,*

$$|\langle \boldsymbol{\theta}_*^i - \tilde{\boldsymbol{\theta}}_t^i, \boldsymbol{x}\rangle| \le (\alpha_t + \beta_t) \cdot \|\boldsymbol{x}\|_{V_t^{-1}}.$$

**Proof:** We first reformulate the expected reward as follows,

$$\langle \boldsymbol{\theta}_*^i, \boldsymbol{x}\rangle = \langle \boldsymbol{\theta}_*^i - \hat{\boldsymbol{\theta}}_t^i, \boldsymbol{x}\rangle + \langle \hat{\boldsymbol{\theta}}_t^i - \tilde{\boldsymbol{\theta}}_t^i, \boldsymbol{x}\rangle + \langle \tilde{\boldsymbol{\theta}}_t^i, \boldsymbol{x}\rangle.$$

Applying the Cauchy-Schwarz inequality (Aldaz et al., 2015), this expression can be bounded as:

$$\langle \boldsymbol{\theta}_*^i, \boldsymbol{x}\rangle - \langle \tilde{\boldsymbol{\theta}}_t^i, \boldsymbol{x}\rangle \le \|\boldsymbol{\theta}_*^i - \hat{\boldsymbol{\theta}}_t^i\|_{V_t}\|\boldsymbol{x}\|_{V_t^{-1}} + \|\hat{\boldsymbol{\theta}}_t^i - \tilde{\boldsymbol{\theta}}_t^i\|_{V_t}\|\boldsymbol{x}\|_{V_t^{-1}}.$$

Since the event $\tilde{\mathcal{E}}$ holds, the inequality can be further relaxed to:

$$\langle \boldsymbol{\theta}_*^i, \boldsymbol{x}\rangle - \langle \tilde{\boldsymbol{\theta}}_t^i, \boldsymbol{x}\rangle \le (\alpha_t + \beta_t) \cdot \|\boldsymbol{x}\|_{V_t^{-1}}.$$

A similar discussion derives that

$$\langle \tilde{\boldsymbol{\theta}}_t^i, \boldsymbol{x}\rangle = \langle \tilde{\boldsymbol{\theta}}_t^i - \hat{\boldsymbol{\theta}}_t^i, \boldsymbol{x}\rangle + \langle \hat{\boldsymbol{\theta}}_t^i - \boldsymbol{\theta}_*^i, \boldsymbol{x}\rangle + \langle \boldsymbol{\theta}_*^i, \boldsymbol{x}\rangle$$

$$\le \|\tilde{\boldsymbol{\theta}}_t^i - \hat{\boldsymbol{\theta}}_t^i\|_{V_t}\|\boldsymbol{x}\|_{V_t^{-1}} + \|\hat{\boldsymbol{\theta}}_t^i - \boldsymbol{\theta}_*^i\|_{V_t}\|\boldsymbol{x}\|_{V_t^{-1}} + \langle \boldsymbol{\theta}_*^i, \boldsymbol{x}\rangle$$

$$\le (\alpha_t + \beta_t) \cdot \|\boldsymbol{x}\|_{V_t^{-1}} + \langle \boldsymbol{\theta}_*^i, \boldsymbol{x}\rangle.$$

Thus, the proof of Lemma 5 is complete. $\square$

In the following, we present two lemmas to analyze the elimination algorithm LAE, which serve as counterparts to Lemma 2 and Lemma 3.

**Lemma 6** *Suppose event $\tilde{\mathcal{E}}$ in Eq. (21) holds. In Algorithm 3, if $\boldsymbol{x}_t^* \in \mathcal{A}_{t,s}$ and $\|\boldsymbol{x}\|_{V_t^{-1}} \le C$ for any $\boldsymbol{x} \in \mathcal{A}_{t,s}$, then*

$$\boldsymbol{x}_t^* \in \mathcal{A}_{t,s}^m \text{ and } \langle \boldsymbol{\theta}_*^i, \boldsymbol{x}_t^* - \boldsymbol{x} \rangle \le 4(1 + \lambda + \cdots + \lambda^{i-1}) \cdot (\alpha_t + \beta_t) \cdot C, \forall i \in [m], \forall \boldsymbol{x} \in \mathcal{A}_{t,s}^m.$$

**Proof:** Similar to the proof of Lemma 2, we prove this lemma by an inductive approach. For the first objective $i = 1$, since $\hat{\boldsymbol{x}}_t^1 = \arg\max_{\boldsymbol{x} \in \mathcal{A}_{t,s}^0} \langle \tilde{\boldsymbol{\theta}}_t^1, \boldsymbol{x} \rangle$ and $\boldsymbol{x}_t^* \in \mathcal{A}_{t,s}^0 = \mathcal{A}_{t,s}$, it follows that for all $\boldsymbol{x} \in \mathcal{A}_{t,s}^1$,

$$\langle \boldsymbol{\theta}_*^1, \boldsymbol{x}_t^* - \boldsymbol{x} \rangle \le \langle \boldsymbol{\theta}_*^1, \boldsymbol{x}_t^* - \boldsymbol{x} \rangle + \langle \tilde{\boldsymbol{\theta}}_t^1, \hat{\boldsymbol{x}}_t^1 - \boldsymbol{x}_t^* \rangle. \tag{22}$$

Given that the event $\tilde{\mathcal{E}}$ holds, Lemma 5 tells that for all $\boldsymbol{x} \in \mathcal{A}_{t,s}^1$,

$$|\langle \boldsymbol{\theta}_*^1 - \tilde{\boldsymbol{\theta}}_t^1, \boldsymbol{x}_t^* \rangle| \le (\alpha_t + \beta_t) \cdot \|\boldsymbol{x}_t^*\|_{V_t^{-1}}, |\langle \boldsymbol{\theta}_*^1 - \tilde{\boldsymbol{\theta}}_t^1, \boldsymbol{x} \rangle| \le (\alpha_t + \beta_t) \cdot \|\boldsymbol{x}\|_{V_t^{-1}}.$$

Substituting this into Eq. (22), it follows that for all $\boldsymbol{x} \in \mathcal{A}_{t,s}^1$,

$$\langle \boldsymbol{\theta}_*^1, \boldsymbol{x}_t^* - \boldsymbol{x} \rangle \le (\alpha_t + \beta_t) \cdot \|\boldsymbol{x}_t^*\|_{V_t^{-1}} + \langle \tilde{\boldsymbol{\theta}}_t^1, \hat{\boldsymbol{x}}_t^1 - \boldsymbol{x} \rangle + (\alpha_t + \beta_t) \cdot \|\boldsymbol{x}\|_{V_t^{-1}}.$$

Recall that $\mathcal{A}_{t,s}^1 = \{\boldsymbol{x} \in \mathcal{A}_{t,s}^0 | \langle \tilde{\boldsymbol{\theta}}_t^1, \hat{\boldsymbol{x}}_t^1 - \boldsymbol{x} \rangle \le 2(\alpha_t + \beta_t) \cdot C\}$. Therefore, for all $\boldsymbol{x} \in \mathcal{A}_{t,s}^1$,

$$\langle \boldsymbol{\theta}_*^1, \boldsymbol{x}_t^* - \boldsymbol{x} \rangle \le (\alpha_t + \beta_t) \cdot \|\boldsymbol{x}_t^*\|_{V_t^{-1}} + 2(\alpha_t + \beta_t) \cdot C + (\alpha_t + \beta_t) \cdot \|\boldsymbol{x}\|_{V_t^{-1}}.$$

Since $\|\boldsymbol{x}\|_{V_t^{-1}} \le C$ for any $\boldsymbol{x} \in \mathcal{A}_{t,s}$, it follows that for all $\boldsymbol{x} \in \mathcal{A}_{t,s}^1$,

$$\langle \boldsymbol{\theta}_*^1, \boldsymbol{x}_t^* - \boldsymbol{x} \rangle \le 4(\alpha_t + \beta_t) \cdot C.$$

Next, Lemma 5 tells that

$$\langle \tilde{\boldsymbol{\theta}}_t^1, \hat{\boldsymbol{x}}_t^1 - \boldsymbol{x}_t^* \rangle \le \langle \boldsymbol{\theta}_*^1, \hat{\boldsymbol{x}}_t^1 - \boldsymbol{x}_t^* \rangle + (\alpha_t + \beta_t) \cdot \|\hat{\boldsymbol{x}}_t^1\|_{V_t^{-1}} + (\alpha_t + \beta_t) \cdot \|\boldsymbol{x}_t^*\|_{V_t^{-1}}.$$

Since $\boldsymbol{x}_t^*$ is the optimal arm, $\langle \boldsymbol{\theta}_*^1, \hat{\boldsymbol{x}}_t^1 - \boldsymbol{x}_t^* \rangle \le 0$. Using $\|\boldsymbol{x}\|_{V_t^{-1}} \le C$ for any $\boldsymbol{x} \in \mathcal{A}_{t,s}$, we have

$$\langle \tilde{\boldsymbol{\theta}}_t^1, \hat{\boldsymbol{x}}_t^1 - \boldsymbol{x}_t^* \rangle \le 2(\alpha_t + \beta_t) \cdot C.$$

Thus, $\boldsymbol{x}_t^* \in \mathcal{A}_t^1 = \{a \in \mathcal{A}_{t,s}^0 | \langle \tilde{\boldsymbol{\theta}}_t^1, \hat{\boldsymbol{x}}_t^1 - \boldsymbol{x}_t^* \rangle \le 2(\alpha_t + \beta_t) \cdot C\}$. The proof for the first objective is finished.

Using the induction method, assume that for $i \ge 2$,

$$\boldsymbol{x}_t^* \in \mathcal{A}_{t,s}^j \text{ and } \langle \boldsymbol{\theta}_*^j, \boldsymbol{x}_t^* - \boldsymbol{x} \rangle \le 4(1 + \lambda + \cdots + \lambda^{j-1}) \cdot (\alpha_t + \beta_t) \cdot C, \forall \boldsymbol{x} \in \mathcal{A}_{t,s}^j, \forall j \in [i-1].$$

We aim to prove

$$\boldsymbol{x}_t^* \in \mathcal{A}_{t,s}^i \text{ and } \langle \boldsymbol{\theta}_*^i, \boldsymbol{x}_t^* - \boldsymbol{x} \rangle \le 4(1 + \lambda + \cdots + \lambda^{i-1}) \cdot (\alpha_t + \beta_t) \cdot C, \forall \boldsymbol{x} \in \mathcal{A}_{t,s}^i. \tag{23}$$

Since $\hat{\boldsymbol{x}}_t^i = \arg\max_{\boldsymbol{x} \in \mathcal{A}_{t,s}^{i-1}} \langle \tilde{\boldsymbol{\theta}}_t^i, \boldsymbol{x} \rangle$ and $\boldsymbol{x}_t^* \in \mathcal{A}_{t,s}^{i-1}$, it follows that for all $\boldsymbol{x} \in \mathcal{A}_{t,s}^i$,

$$\langle \boldsymbol{\theta}_*^i, \boldsymbol{x}_t^* - \boldsymbol{x} \rangle \le \langle \boldsymbol{\theta}_*^i, \boldsymbol{x}_t^* - \boldsymbol{x} \rangle + \langle \tilde{\boldsymbol{\theta}}_t^i, \hat{\boldsymbol{x}}_t^i - \boldsymbol{x}_t^* \rangle. \tag{24}$$

Given that the event $\tilde{\mathcal{E}}$ holds, Lemma 5 tells ensures that for all $\boldsymbol{x} \in \mathcal{A}_{t,s}^i$,

$$|\langle \boldsymbol{\theta}_*^i - \tilde{\boldsymbol{\theta}}_t^i, \boldsymbol{x}_t^* \rangle| \le (\alpha_t + \beta_t) \cdot \|\boldsymbol{x}_t^*\|_{V_t^{-1}}, |\langle \boldsymbol{\theta}_*^i - \tilde{\boldsymbol{\theta}}_t^i, \boldsymbol{x} \rangle| \le (\alpha_t + \beta_t) \cdot \|\boldsymbol{x}\|_{V_t^{-1}}.$$

Substituting this into Eq. (24), for all $\boldsymbol{x} \in \mathcal{A}_{t,s}^i$,

$$\langle \boldsymbol{\theta}_*^i, \boldsymbol{x}_t^* - \boldsymbol{x} \rangle \le (\alpha_t + \beta_t) \cdot \|\boldsymbol{x}_t^*\|_{V_t^{-1}} + \langle \tilde{\boldsymbol{\theta}}_t^i, \hat{\boldsymbol{x}}_t^i - \boldsymbol{x} \rangle + (\alpha_t + \beta_t) \cdot \|\boldsymbol{x}\|_{V_t^{-1}}.$$

Recalling that $\mathcal{A}_{t,s}^i = \{\boldsymbol{x} \in \mathcal{A}_{t,s}^{i-1} | \langle \tilde{\boldsymbol{\theta}}_t^i, \hat{\boldsymbol{x}}_t^i - \boldsymbol{x} \rangle \le (2 + 4\lambda + \cdots + 4\lambda^{i-1}) \cdot (\alpha_t + \beta_t) \cdot C\}$ and $\|\boldsymbol{x}\|_{V_t^{-1}} \le C$ for any $\boldsymbol{x} \in \mathcal{A}_{t,s}$, we have for all $\boldsymbol{x} \in \mathcal{A}_{t,s}^i$,

$$\langle \boldsymbol{\theta}_*^i, \boldsymbol{x}_t^* - \boldsymbol{x} \rangle \le 4(1 + \lambda + \cdots + \lambda^{i-1}) \cdot (\alpha_t + \beta_t) \cdot C.$$

Finally, Lemma 5 tells that

$$\langle \tilde{\boldsymbol{\theta}}_t^i, \hat{\boldsymbol{x}}_t^i - \boldsymbol{x}_t^* \rangle \leq \langle \boldsymbol{\theta}_*^i, \hat{\boldsymbol{x}}_t^i - \boldsymbol{x}_t^* \rangle + 2(\alpha_t + \beta_t) \cdot C.$$

Using Eq. (3), $\langle \boldsymbol{\theta}_*^i, \hat{\boldsymbol{x}}_t^i - \boldsymbol{x}_t^* \rangle \leq \lambda \cdot \max_{j \in [i-1]} \{\langle \boldsymbol{\theta}_*^j, \boldsymbol{x}_t^* - \hat{\boldsymbol{x}}_t^i \rangle\}$. Thus,

$$\langle \tilde{\boldsymbol{\theta}}_t^i, \hat{\boldsymbol{x}}_t^i - \boldsymbol{x}_t^* \rangle \leq \lambda \cdot 4(1 + \lambda + \cdots + \lambda^{i-2}) \cdot (\alpha_t + \beta_t) \cdot C + 2(\alpha_t + \beta_t) \cdot C.$$

It follows that $\boldsymbol{x}_t^* \in \mathcal{A}_{t,s}^i = \{\boldsymbol{x} \in \mathcal{A}_{t,s}^{i-1} | \langle \tilde{\boldsymbol{\theta}}_t^i, \hat{\boldsymbol{x}}_t^i - \boldsymbol{x}_t^* \rangle \leq (2 + 4\lambda + \cdots + \lambda^{i-1}) \cdot (\alpha_t + \beta_t) \cdot C\}$. This completes the proof of Eq. (23) and concludes the induction framework. $\square$

Lemma 6 depends on the assumption that $\boldsymbol{x}_t^* \in \mathcal{A}_{t,s}^0$. In the following lemma, we remove this assumption.

**Lemma 7** *Suppose $\tilde{\mathcal{E}}$ in Eq. (21) holds. In Algorithm 2, for any $s \geq 1$ and $\boldsymbol{x} \in \mathcal{A}_{t,s}$,*

$$\langle \boldsymbol{\theta}_*^i, \boldsymbol{x}_t^* - \boldsymbol{x} \rangle \leq 4(1 + \lambda + \cdots + \lambda^{i-1}) \cdot (\alpha_t + \beta_t) \cdot 2^{-s+1}, i \in [m].$$

**Proof:** We prove $\boldsymbol{x}_t^* \in \mathcal{A}_{t,s}^0$ for $s \geq 1$ by induction. For the base case $s = 1$, $\boldsymbol{x}_t^* \in \mathcal{A}_{t,1}^0$ obviously since $\mathcal{A}_{t,1}^0 = \mathcal{A}_t$. Assume that $\boldsymbol{x}_t^* \in \mathcal{A}_{,s}^0$ for some $s \geq 1$. By Lemma 6, $\boldsymbol{x}_t^* \in \mathcal{A}_{t,s}^0$ deduces that $\boldsymbol{x}_t^* \in \mathcal{A}_{t,s}^m$. Since $\mathcal{A}_{t,s+1}^0 = \mathcal{A}_{t,s+1} = \mathcal{A}_{t,s}^m$, it follows that $\boldsymbol{x}_t^* \in \mathcal{A}_{t,s+1}^0$. By induction, we conclude that $\boldsymbol{x}_t^* \in \mathcal{A}_{t,s}^0$ for all $s \geq 1$. Given $\boldsymbol{x}_t^* \in \mathcal{A}_{t,s}^0$, Lemma 6 tells that for any $\boldsymbol{x} \in \mathcal{A}_{t,s}^m$, $\langle \boldsymbol{\theta}_*^i, \boldsymbol{x}_t^* - \boldsymbol{x} \rangle \leq 4(1 + \lambda + \cdots + \lambda^{i-1}) \cdot 2^{-s}, i \in [m]$. Thus, Lemma 7 holds since $\mathcal{A}_{t,s}^m = \mathcal{A}_{t,s+1}$. $\square$

By a similar argument as in Lemma 7, we obtain the following lemma.

**Lemma 8** *Suppose $\tilde{\mathcal{E}}$ in Eq. (21) holds. In Algorithm 2, for any $\boldsymbol{x} \in \mathcal{A}_{t,T}$,*

$$\langle \boldsymbol{\theta}_*^i, \boldsymbol{x}_t^* - \boldsymbol{x} \rangle \leq 4(1 + \lambda + \cdots + \lambda^{i-1}) \cdot (\alpha_t + \beta_t) \cdot \frac{1}{\sqrt{T}}, i \in [m].$$

We now complete the proof of Theorem 2. Let $\psi_s(T) = \{t \in [T] | \|\boldsymbol{x}_t\|_{V_t^{-1}} > 2^{-s}\}$ for $s \geq 1$, and let $\psi_0(T) = \{t \in [T] | \|\boldsymbol{x}_t\|_{V_t^{-1}} \leq 1/\sqrt{T}\}$. The regret can be decomposed as

$$R^i(T) = \sum_{t \in \psi_0(T)} \langle \boldsymbol{\theta}_*^i, \boldsymbol{x}_t^* - \boldsymbol{x}_t \rangle + \sum_{s=1}^{S} \sum_{t \in \psi_s(T)} \langle \boldsymbol{\theta}_*^i, \boldsymbol{x}_t^* - \boldsymbol{x}_t \rangle.$$

where $S \leq \log(T)$, since $2^{-\log T} \leq 1/\sqrt{T}$.

The trials in $\psi_0(T)$ play arms in the **if** case. By Lemma 8, we have

$$\sum_{t \in \psi_0(T)} \langle \boldsymbol{\theta}_*^i, \boldsymbol{x}_t^* - \boldsymbol{x}_t \rangle \leq |\psi_0(T)| \cdot 4(1 + \lambda + \cdots + \lambda^{i-1}) \cdot (\alpha_T + \beta_T) \cdot \frac{1}{\sqrt{T}}.$$

For the trials in $\psi_s(T)$, corresponding to the **else if** case, where the arm is selected from $\mathcal{A}_{t,s}$. Lemma 7 tells that

$$\sum_{t \in \psi_s(T)} \langle \boldsymbol{\theta}_*^i, \boldsymbol{x}_t^* - \boldsymbol{x}_t \rangle \leq |\psi_s(T)| \cdot 4(1 + \lambda + \cdots + \lambda^{i-1}) \cdot (\alpha_T + \beta_T) \cdot 2^{-s+1}.$$

Thus, the regret for the $i$-the objective is bounded as

$$R^i(T) \leq 4(1 + \lambda + \cdots + \lambda^{i-1}) \cdot (\alpha_T + \beta_T) \cdot \left( \frac{|\psi_0(T)|}{\sqrt{T}} + \sum_{s=1}^{S} 2 \cdot 2^{-s} |\psi_s(T)| \right). \tag{25}$$

Lemma 3 of Chu et al. (2011) states that

$$\sum_{t \in \psi_s(T)} \|\boldsymbol{x}_t\|_{V_t^{-1}} \leq 5\sqrt{d|\psi_s(T)| \log(|\psi_s(T)|)}.$$

Using the fact that $\|\boldsymbol{x}_t\|_{V_t^{-1}} > 2^{-s}$ for $t \in \psi_s(T)$, we obtain

$$2^{-s}|\psi_s(T)| \le 5\sqrt{d|\psi_s(T)|\log(|\psi_s(T)|)}.$$

Since $|\psi_s(T)| \le T$, we obtain

$$\sum_{s=1}^{S} 2 \cdot 2^{-s}|\psi_s(T)| \le 10\sum_{s=1}^{S} \sqrt{d|\psi_s(T)|\log T}.$$

Applying the Cauchy-Schwarz inequality (Aldaz et al., 2015), this simplifies to

$$\sum_{s=1}^{S} 2 \cdot 2^{-s}|\psi_s(T)| \le 10\gamma_T\sqrt{dST\log T}.$$

Since $S \le \log T$, we can further relax this bound to

$$\sum_{s=1}^{S} 2 \cdot 2^{-s}|\psi_s(T)| \le 10\log T\sqrt{dT}.$$

Substituting this result into Eq. (25) shows that

$$R^i(T) \le 4(1 + \lambda + \cdots + \lambda^{i-1}) \cdot (\alpha_T + \beta_T) \cdot \left(\sqrt{T} + 10\log T\sqrt{dT}\right).$$

A simple relaxation yields the final bound,

$$R^i(T) \le 44(1 + \lambda + \cdots + \lambda^{i-1}) \cdot (\alpha_T + \beta_T) \cdot \log T\sqrt{dT}.$$

The proof of Theorem 2 is finished. $\qquad\square$

