# OpenReview forum: "Multi-Objective Bandits with Hierarchical Preferences: A Thompson Sampling Approach"
_ICLR.cc/2026/Conference — ICLR 2026 Conference Withdrawn Submission_

### Official Review · Reviewer_e4BD · 2025-10-25

**Soundness:** 3
**Presentation:** 2
**Contribution:** 2
**Rating:** 6
**Confidence:** 4

**Summary:**

This paper introduces Thompson Sampling–based algorithms for multi-objective bandits with hierarchical (lexicographic) preferences. It proposes two algorithms. When the distribution type is known and is Gaussian, DK-TSLB algorithm achieves logarithmic gap-dependent regret. The other algorithm (DF-TSLB) works under the setting of stochastic linear bandits and achieves $\sqrt{T}$ gap-independent regret. The theoretical analysis proves that these bounds are comparable to single-objective TS algorithms, except for an additional multiplicative factor that quantifies hierarchical trade-offs. Simulations show the advantage of the proposed algorithms with respect to baseline methods.

**Strengths:**

1. To the best of the reviewer’s knowledge, this is the first TS-based solution for lexicographic multi-objective bandits. The results extend the reach of TS into multi-objective decision-making with lexicographic preferences, which is becoming highly relevant for real-world applications (recommender systems, clinical trials, etc).

2. The paper provides a detailed theoretical analysis with complete proofs. Both problem-dependent and minimax regret bounds are established.

3. The paper is easy to follow. Assumptions and algorithms are clearly described. The results make sense.

**Weaknesses:**

1. The hierarchical tradeoff parameterized by $\lambda$ restricts the set of multi-objective learning problems to which the proposed methods are applicable. The hierarchical tradeoff may not hold in practice. This requires empirical justification (with some well-known datasets used in bandit algorithm evaluation) to improve the credibility of the current work.

2. Experiments are limited to toy settings (10 arms, 3 objectives). No evaluation on real-world datasets.

3. Dependence of the regret on the additional multiplicative factor $\Lambda_i(\lambda)$ is an unresolved question. A lower bound analysis is not provided in the paper.

4. The paper does not compare against more recent lexicographic multi-objective methods beyond PF-LEX [A, B]. These works also involve the hierarchical tradeoff parameter $\lambda$. A detailed comparison in terms of algorithm implementation, regret bounds, and proof techniques should be given. Specifically, the filtering approach in these works should be compared with the filtering approach of the TS-based algorithms (conceptually, what is new in the paper?). The novelty and challenges of extending the results to the TS-based algorithms should be articulated clearly to evaluate the originality and significance of the proposed approach.

[A] Xue, Bo, et al. "Multiobjective Lipschitz bandits under lexicographic ordering." Proceedings of the AAAI Conference on Artificial Intelligence. Vol. 38. No. 15. 2024.

[B]  Xue, Bo, et al. "Multiple trade-offs: An improved approach for lexicographic linear bandits." Proceedings of the AAAI Conference on Artificial Intelligence. Vol. 39. No. 20. 2025.

**Questions:**

1. Indicate what the summation is over in the regret bound in line 20 of the abstract.

2. Algorithms require the knowledge of the tradeoff parameter $\lambda$. What happens if $\lambda$ is misspecified? An experimental evaluation of the proposed algorithms under misspecification will yield an understanding of the robustness of the proposed algorithms.

3. Can $\lambda$ be estimated online? An experimental evaluation of adaptations of algorithms that can estimate $\lambda$ online will strengthen the contribution.

---

> ### Author Response · Authors · 2025-11-17
> **Response to Reviewer e4BD: Part I**
>
> We sincerely appreciate the constructive feedback and have thoroughly evaluated the raised concerns. Our point-by-point responses to the weaknesses and questions follow below.
>
> ---
>
> **W1.** *The hierarchical tradeoff parameterized by $\lambda$ restricts the set of multi-objective learning problems to which the proposed methods are applicable. The hierarchical tradeoff may not hold in practice. This requires empirical justification (with some well-known datasets used in bandit algorithm evaluation) to improve the credibility of the current work.*
>
> **R1.** $\lambda$ restricts the problem to a well-defined class of lexicographic bandit instances where rigorous theoretical analysis is tractable. Since no single algorithm can efficiently handle all possible multi-objective problems, assumptions (2) and (3) provide a principled way to capture the trade-offs between objectives and allow us to derive algorithms with provable regret guarantees.
>
> It is also worth noting that our algorithm does not require the exact value of $\lambda$. Any sufficiently large upper bound of $\lambda$ that satisfies Eq. (2) or Eq. (3) is enough to guarantee sublinear regret with respect to $T$. As a result, the algorithm is robust and performs well across a broad range of $\lambda$ values.
>
> ---
>
> **W2.** *Experiments are limited to toy settings (10 arms, 3 objectives). No evaluation on real-world datasets.*
>
> **R2.** We appreciate the reviewer’s concern. Existing studies on lexicographic bandits, such as [A], [B], and [C], also conduct evaluations in synthetic settings with 10–20 arms and 2–3 objectives. This is largely due to the mismatch between lexicographic preference structures and available real-world bandit datasets. We are actively searching for or constructing suitable datasets to address this gap.
>
> *Reference.*
> [A] Xue, Bo, et al. "Multiobjective Lipschitz bandits under lexicographic ordering." Proceedings of the AAAI Conference on Artificial Intelligence. Vol. 38. No. 15. 2024.
> [B] Xue, Bo, et al. "Multiple trade-offs: An improved approach for lexicographic linear bandits." Proceedings of the AAAI Conference on Artificial Intelligence. Vol. 39. No. 20. 2025.
> [C] Huyuk et al. "Multi-objective multi-armed bandit with lexicographically ordered and satisficing objectives." Machine Learning, 110(6):1233–1266, 2021.
>
> ---
>
> **W3.** *Dependence of the regret on the additional multiplicative factor $\Lambda ^i(\lambda)$ is an unresolved question. A lower bound analysis is not provided in the paper.*
>
> **R3.** The exponential growth of $\Lambda^i(\lambda)$ is fundamentally rooted in the nature of lexicographic optimization in multi-dimensional spaces. Each additional objective introduces a new dimension of trade-offs that must be considered while maintaining the strict priority ordering. Consequently, the **search space of identifying lex-optimal solutions** increases exponentially with the number of objectives, and our regret bounds accurately reflect this fundamental challenge. We emphasize two crucial points about this dependence:
>
> - The primary objective ($i=1$) maintains $\Lambda^1(\lambda)=1$ for all $\lambda \geq 0$, meaning its performance matches existing single-objective bandit algorithms regardless of the number of secondary objectives.
> - Many real-world multi-objective problems naturally have a small number of truly critical objectives ($m \leq 3 \text{ or }5$), where the exponential term remains manageable.
>
> We acknowledge that this dependence may limit scalability to problems with very large numbers of objectives. However, for the important class of problems with a few prioritized objectives, our approach provides the first solution with strict priority guarantees.

---

> ### Author Response · Authors · 2025-11-17
> **Response to Reviewer e4BD: Part II**
>
> **W4.** *The paper does not compare against more recent lexicographic multi-objective methods beyond PF-LEX [A, B]. These works also involve the hierarchical tradeoff parameter $\lambda$. A detailed comparison in terms of algorithm implementation, regret bounds, and proof techniques should be given. Specifically, the filtering approach in these works should be compared with the filtering approach of the TS-based algorithms (conceptually, what is new in the paper?). The novelty and challenges of extending the results to the TS-based algorithms should be articulated clearly to evaluate the originality and significance of the proposed approach.*
>
> **A4.** Thank you for highlighting this point. **In the revised Section 5**, we have added a detailed comparison with the recent lexicographic multi-objective works [A] and [B], covering algorithmic design, regret guarantees, and proof techniques. Specifically:
>
> - *Algorithmic design.* In Algorithm 1, we can **directly select the arm with the largest posterior uncertainty** (Step 5), which is possible because the arm set is finite. In contrast, both [A] and [B] rely on a **repeat-until search procedure** to locate a sufficiently uncertain arm in a continuous or high-dimensional space. Thus, although all works use $\lambda$-based hierarchical filtering, the exploration mechanisms are fundamentally different.
> - *Regret guarantees.* Algorithm 1 provides an **instance-dependent** regret bound that explicitly reflects the hardness parameters $\Delta_i(a)$. In comparison, [A] and [B] derive **instance-independent** bounds, as their settings and techniques do not naturally yield gap-dependent analysis.
> - *Proof techniques and conceptual novelty.* The main contribution of our work is demonstrating that **BayesUCB-/TS-style posterior-driven exploration** can be combined with $\lambda$-based hierarchical elimination to solve lexicographic bandit problems, which differs from the deterministic UCB-style analyses in [A] and [B].
>
> ---
>
> **Q1.** *Indicate what the summation is over in the regret bound in line 20 of the abstract.*
>
> **A1.** The summation is over all arms with positive gap $\Delta^i(a) > 0$, i.e., terms of the form $\sum_{\Delta^i(a)>0} \frac{\log T}{\Delta^i(a)}.$ We have clarified this explicitly in the revised abstract.
>
> ---
>
> **Q2.** *Algorithms require the knowledge of the tradeoff parameter$\lambda$. What happens if $\lambda$ is misspecified?*
>
> **A2.** Our algorithms do not require the *exact* value of $\lambda$. Any sufficiently large **upper bound** of $\lambda$ that satisfies Eq. (2) or Eq. (3) is enough to ensure sublinear regret in $T$. Therefore, the algorithms are inherently robust to misspecification: as long as $\lambda$ upper-bounds the true trade-off parameter, the regret guarantees remain valid and the empirical performance remains stable.
>
> However, if $\lambda$ does not satisfy Eq. (2) or Eq. (3), for example if it underestimates the true trade-off strength, the filtering procedure may incorrectly eliminate the lexicographic optimal arm. In this case, the algorithm may suffer linear regret.
>
> ---
>
> **Q3.** *Can $\lambda$ be estimated online? An experimental evaluation of adaptations of algorithms that can estimate $\lambda$ online will strengthen the contribution.*
>
> **A3.** At this stage, we do not have an online procedure for estimating $\lambda$ with theoretical guarantees. Designing such a method is challenging because using empirical mean rewards to estimate $\lambda$ is inherently unstable: the empirical mean differences appear in both the numerator and denominator of the definition, which can easily lead to unbounded or highly noisy estimates.
>
> We view developing a theoretically sound online estimator of $\lambda$ as an important and nontrivial open direction, and we are actively working on this problem.
>
> ---
>
> Thank you again for your time and constructive feedback. We are happy to further discuss any aspect of our work.

---

> > ### Comment · Reviewer_e4BD · 2025-11-23
> >
> > Thank your for the response, which partly address my questions. For now, I would like to keep my rating.

---

> > > ### Author Response · Authors · 2025-11-24
> > > **Many Thanks for the Follow-up**
> > >
> > > Thank you very much for your follow-up. We fully respect your decision to keep the current rating.
> > >
> > > If there are any aspects where further clarification could be useful or if additional evidence or analysis would help address the remaining concerns, we would be very glad to provide it.
> > >
> > > We genuinely appreciate your time and thoughtful engagement with our work, and we are happy to assist in any way that could support your evaluation.

---

### Official Review · Reviewer_W7Sg · 2025-10-27

**Soundness:** 2
**Presentation:** 2
**Contribution:** 1
**Rating:** 2
**Confidence:** 3

**Summary:**

This paper proposes and analyzes Thompson sampling-like algorithms for multi-objective bandits with hierarchical preferences (lexicographic ordering). Two distinct settings are investigated: (1) MOMAB: Finite, independent arms with Gaussian rewards and Gaussian priors, where the distribution parameters are known up to the prior. (2) MOSLB: Stochastic linear bandits with shared parameters across arms and general bounded reward distributions, where distributions are unknown. The authors present Algorithm 1 (DK-TSLB) for the MOMAB setting and Algorithm 2 (DF-TSLB) for the MOSLB setting. Both algorithms employ a priority-based arm elimination procedure. Algorithm 1 builds upon Bayesian inference, while Algorithm 2 utilizes TS-like posterior sampling. A key assumption is that suboptimal expected rewards in higher-priority objectives ($j<i$) can lead to at most $\lambda$-times the improvement in lower-priority objectives ($i$). Under this structure, the authors derive an instance-dependent regret bound for Algorithm 1 and an instance-independent regret bound for Algorithm 2. Both algorithms achieve sublinear regret for each objective $i \in [m]$. Notably, the regret guarantee for the primary objective ($i=1$) is shown to be no worse than standard single-objective bandit algorithms.

**Strengths:**

The concept of using the parameter $\lambda$ to model the trade-off structure between hierarchically ordered objectives is clever, simplifying both the problem setup and the subsequent analysis. It is an interesting theoretical finding that, under the proposed framework, optimizing for lower-priority objectives ($i > 1$) does not degrade the regret performance for the primary objective ($i=1$) compared to single-objective optimal algorithms.

**Weaknesses:**

1. **Role and Definition of Thompson Sampling**: The paper frames its contribution around Thompson Sampling, but TS doesn't appear to be the core mechanism driving the results. The arm elimination procedure, which manages exploration across objectives based on the $\lambda$-structure, seems far more critical.

- Algorithm 1 (DK-TSLB) does not implement the characteristic posterior sampling step of TS; it uses Bayesian inference (calculating posterior means and variances) combined with confidence bounds for elimination. It would be more accurately described as a Bayesian UCB or Active Arm Elimination algorithm.

- While Algorithm 2 (DF-TSLB) does incorporate posterior sampling ($\tilde{\theta}$), it's unclear from the presentation and analysis whether this sampling step is truly essential for achieving the regret bound, or if using the posterior mean ($\hat{\theta}$) would suffice (related to Question 1).

2. **Novelty Relative to Prior Work**: Given that TS itself might not be the key innovation, the main contribution appears to be the application of the $\lambda$-based hierarchical elimination procedure to MOMAB and MOSLB settings. However, a very similar structure (Assumption 5) and elimination logic (Algorithms 2 & 4 involving $\lambda_i = 1+\lambda+...+\lambda^{i-1}$) was previously introduced in Xue et al. (2024), "Multiobjective Lipschitz Bandits under Lexicographic Ordering". While cited, the current paper does not clearly state the extent to which it adopts this existing mechanism, making the novelty seem incremental rather than substantial.

3. **Disparate Settings and Analyses**: The two settings investigated (MOMAB with known distributions vs. MOSLB with unknown distributions) feel somewhat disconnected.
- The models differ significantly (independent arms vs. shared linear parameters).
- The algorithms employ different core mechanics (Bayesian UCB vs. TS with sampling).
- The resulting regret bounds are of different types (instance-dependent vs. instance-independent).
- This lack of alignment makes the paper feel like a combination of two separate results rather than a unified contribution built around a single core idea (beyond the shared $\lambda$-elimination concept).

4. **Exponential Dependence on Objective Priority**: Although acknowledged briefly by the authors as an open problem 2, the exponential dependence of the regret bounds on the objective index $i$ (via the $\Lambda^i(\lambda)$ term) is a significant limitation. This could render the theoretical guarantees impractical for problems with even a moderate number of objectives ($m$) or a large trade-off factor ($\lambda$). A more thorough discussion and analysis of this limitation is warranted.

**Questions:**

1. **Necessity of Sampling in Algorithm 2**: Could the authors clarify why posterior sampling ($\tilde{\theta}$) is necessary in Algorithm 2? Specifically, what would happen if $\hat{\theta}_t^i$ (the posterior mean) were used instead of $\tilde{\theta}_t^i$ for arm elimination (in Algorithm 3)? Would the algorithm fail, or would the proof break down? If the proof fails, which specific step or lemma becomes invalid?

2. **Choice of Regret Analysis**: Why was an instance-dependent regret analysis performed for MOMAB (Thm 1) but an instance-independent one for MOSLB (Thm 2)? Could a problem-independent bound be derived for Algorithm 1, or a problem-dependent bound for Algorithm 2, to allow for a more consistent comparison or demonstrate robustness across analysis types?

3. **Relation to Xue et al. (2024) and Justification of $\lambda$**: Beyond the difference in bandit models (MAB/SLB vs. Lipschitz), what is the key conceptual difference in the $\lambda$-based assumption and the arm elimination scheme compared to Xue et al. (2024)? Could the authors provide a stronger justification for the $\lambda$ assumption itself? Is it purely an analytical tool, or does it reflect structures found in real-world lexicographic problems? How easily can $\lambda$ be estimated or bounded in practice?

4. **Empirical Verification of Scaling**: Does the empirical performance shown in the experiments (Figure 1) reflect the exponential scaling with the objective index $i$ suggested by the $\Lambda^i(\lambda)$ term in the theoretical regret bounds, particularly for the second ($i=2$) and third ($i=3$) objectives?

---

> ### Author Response · Authors · 2025-11-17
> **Response to Reviewer W7Sg: Part I**
>
> We sincerely appreciate the constructive feedback and have thoroughly evaluated the raised concerns. Our point-by-point responses to the weaknesses and questions follow below.
>
> ---
>
> **W1.** *Algorithm 1 (DK-TSLB) does not implement the characteristic posterior sampling step of TS; it uses Bayesian inference (calculating posterior means and variances) combined with confidence bounds for elimination. It would be more accurately described as a Bayesian UCB or Active Arm Elimination algorithm.*
>
> **R1.** We thank the reviewer for pointing this out. You are correct that Algorithm 1 aligns more closely with the Bayesian UCB approach of [Kaufmann et al., 2012a] rather than classical Thompson Sampling. We have updated the terminology accordingly in the revised version of the paper.  Algorithm 1 is now referred to as **“Distribution-Known BayesUCB for Lexicographic Bandits (DK-BULB)”**, and all corresponding expressions have been updated accordingly.
>
> ---
>
> **W2.** *Given that TS itself might not be the key innovation, the main contribution appears to be the application of the $\lambda$-based hierarchical elimination procedure to MOMAB and MOSLB settings. However, a very similar structure (Assumption 5) and elimination logic (Algorithms 2 $\&$ 4 involving $\Lambda^i(\lambda)=1+\lambda+\cdots+\lambda^{i-1}$) was previously introduced in Xue et al. (2024), "Multiobjective Lipschitz Bandits under Lexicographic Ordering". While cited, the current paper does not clearly state the extent to which it adopts this existing mechanism, making the novelty seem incremental rather than substantial.*
>
> **R2.** Thank you for the observation. In the revised Section 5, we have clarified that the $\lambda$-based hierarchical elimination originates from the prior work [Xue et al., 2024]. One of our key contributions is to demonstrate that a Bayesian posterior-based method can be effectively combined with this elimination mechanism to handle lexicographic MOMAB and MOSLB. This integration is not addressed in the earlier Lipschitz-bandit setting.
>
> ---
>
> **W3.** *The two settings investigated (MOMAB with known distributions vs. MOSLB with unknown distributions) feel somewhat disconnected.*
>
> **R3.** The distinction between the two settings is intentional. Our goal is to illustrate how to design different lexicographic multi-objective bandit algorithms under **different structural assumptions**. In MOMAB, the arm set is fixed and finite, which allows us to directly select the arm with the largest posterior uncertainty (Step 5 of Algorithm 1). In contrast, MOSLB features an infinite and potentially changing arm set, requiring a **repeat-until** search procedure to identify the most uncertain arm. Presenting both cases highlights how the algorithmic design and regret analysis differ across these fundamentally different models.
>
> ---
>
> **W4.** *Although acknowledged briefly by the authors as an open problem 2, the exponential dependence of the regret bounds on the objective index $i$ (via the $\Lambda^i(\lambda)$ term) is a significant limitation. This could render the theoretical guarantees impractical for problems with even a moderate number of objectives $m$ or a large trade-off factor $\lambda$. A more thorough discussion and analysis of this limitation is warranted.*
>
> **R4.** The exponential growth of $\Lambda^i(\lambda)$ is fundamentally rooted in the nature of lexicographic optimization in multi-dimensional spaces. Each additional objective introduces a new dimension of trade-offs that must be considered while maintaining the strict priority ordering. Consequently, **the search space of identifying lex-optimal solutions increases exponentially with the number of objectives**, and our regret bounds accurately reflect this fundamental challenge. We emphasize two crucial points about this dependence:
>
> - The primary objective ($i=1$) maintains $\Lambda^1(\lambda)=1$ for all $\lambda \geq 0$, meaning its performance matches existing single-objective bandit algorithms regardless of the number of secondary objectives.
> - Many real-world multi-objective problems naturally have a small number of truly critical objectives ($m \leq 3 \text{ or }5$), where the exponential term remains manageable.
>
> We acknowledge that this dependence may limit scalability to problems with very large numbers of objectives. However, for the important class of problems with a few prioritized objectives, our approach provides the first Bayesian solution with strict priority guarantees.

---

> ### Author Response · Authors · 2025-11-17
> **Response to Reviewer W7Sg: Part II**
>
> **Q1.** *Could the authors clarify why posterior sampling ($\tilde{\theta}$) is necessary in Algorithm 2? Specifically, what would happen if $\hat{\theta}_t^i$ (the posterior mean) were used instead of $\tilde{\theta}_t^i$ for arm elimination (in Algorithm 3)? Would the algorithm fail, or would the proof break down? If the proof fails, which specific step or lemma becomes invalid?*
>
> **A1.** Thank you for this important question. After re-checking the algorithm and proofs, we conclude that if we replace the posterior *sample* $\tilde{\theta}_t^i$ by the posterior *mean* $\hat{\theta}_t^i$ in Algorithm 3 (LAE), the elimination rules still make sense: the posterior mean is a valid point estimate, and with suitable confidence bounds one can carry out lexicographic elimination using $\hat{\theta}_t^i$. However, that replacement turns the method into a deterministic, Bayesian-UCB-style algorithm rather than a Thompson-sampling-style procedure.
>
> By contrast, keeping $\tilde{\theta}_t^i$ lets us present an alternative Thompson-sampling-style analysis. The sampling viewpoint offers different intuition and proof techniques from previous techniques that classify arms as saturated and unsaturated (Agrawal $\&$ Goyal, 2013a) or utilize the supporting functions (Abeille $\&$ Lazaric, 2017).
>
> ---
>
> **Q2.** *Why was an instance-dependent regret analysis performed for MOMAB (Thm 1) but an instance-independent one for MOSLB (Thm 2)? Could a problem-independent bound be derived for Algorithm 1, or a problem-dependent bound for Algorithm 2, to allow for a more consistent comparison or demonstrate robustness across analysis types?*
>
> **A2.** Thank you for this excellent question. We used different types of regret bounds deliberately to highlight how the two models and analysis techniques differ. Concretely:
>
> - Our instance-dependent bound for **MOMAB** (Theorem 1) emphasizes how the gaps $\Delta_i(a)$ control difficulty in the finite-arm setting.
> - Using standard conversions from gap-based bounds to worst-case bounds (e.g. by replacing $\sum_{\Delta^i(a)>0}\frac{\log T}{\Delta^i(a)}$ with $\tilde{O}(\sqrt{KT})$), Algorithm 1 also admits a **problem-independent** bound of the form $\widetilde{O}(\Lambda^i(\lambda)\sqrt{KT})$.
> - For **MOSLB** (Algorithm 2), we provided a problem-independent bound (Theorem 2) because the infinite arm setting and our proof technique naturally yield a $d^{3/2}\sqrt{T}$-type rate. Deriving a meaningful **instance-dependent** bound for Algorithm 2 is substantially more challenging in the infinite-armed linear setting; we have not obtained such a bound yet and therefore treat it as an interesting open problem. We have stated this explicitly in the revision.
>
> ---
>
> **Q3.** *Beyond the difference in bandit models (MAB/SLB vs. Lipschitz), what is the key conceptual difference in the$\lambda$-based assumption and the arm elimination scheme compared to Xue et al. (2024)?*
>
> **A3.** Beyond the different bandit models, the key distinction is how the $\lambda$-based assumption interacts with arm selection. In our MOMAB setting (Algorithm 1), the arm set is finite, so we can **directly choose the arm with the largest posterior uncertainty**, leading to a purely posterior-driven elimination process. In Xue et al. (2024), the continuous arm space requires a **repeat-until search** to locate an informative arm, so the $\lambda$-based elimination operates in a fundamentally different way.
>
> ---
>
> **Q4.** *Could the authors provide a stronger justification for the $\lambda$ assumption itself? Is it purely an analytical tool, or does it reflect structures found in real-world lexicographic problems?*
>
> **A4.** The parameter $\lambda$ is introduced to quantify the difficulty of identifying the optimal arm. Consider three arms with expected rewards: $[3, 0, 3]$ (optimal), $[3, -1, 6]$, and $[2.5, 5, -1]$. The challenges become apparent since: (1) first objective values are too close to distinguish (3 vs 3 vs 2.5); (2) a naive UCB approach on the second objective might wrongly select $[2.5,5,-1]$ due to its large second-objective reward, despite being suboptimal lexicographically.
>
> A large $\lambda$ captures such scenarios, indicating there exist arms with substantially better rewards for objective $i$ while being only slightly worse for previous $i-1$ objectives, thereby complicating the identification of the optimal arm.

---

> ### Author Response · Authors · 2025-11-17
> **Response to Reviewer W7Sg: Part III**
>
> **Q5.** *How easily can $\lambda$ be estimated or bounded in practice?*
>
> **A5.** The practical challenge lies in estimating a suitable value of $\lambda$, which can often be informed by domain-specific expert knowledge. For example, in clinical IMRT planning [1], a slight relaxation (2-3$\%$) in the primary objective (PTV coverage) can lead to over 15$\%$ improvement in secondary objectives (e.g., OAR doses), implying $\lambda$ values in the range of 5–7 . In traffic signal control [2], a 30$\%$ reduction in pedestrian delay may correspond to only a 10$\%$ improvement in vehicle emissions, suggesting $\lambda \approx 0.3$.
>
> It is also worth noting that our algorithm does not require the exact value of $\lambda$. Any sufficiently large upper bound $\lambda$ that satisfies Eq. (2) or Eq. (3) is enough to guarantee sublinear regret with respect to $T$. As a result, the algorithm is robust and performs well across a broad range of $\lambda$ values.
>
>
>
> *Reference.*
> [1] Lexicographic ordering: intuitive multicriteria optimization for IMRT. Physics in Medicine $\&$ Biology, 2007
> [2] Multi-objective Optimization Framework for Trade‑Off Among Pedestrian Delays and Vehicular Emissions at Signal‑Controlled Intersections. Arabian Journal for Science and Engineering, 2024.
>
> ---
>
> **Q6.** *Does the empirical performance shown in the experiments (Figure 1) reflect the exponential scaling with the objective index $i$ suggested by the $\Lambda^i(\lambda)$ term in the theoretical regret bounds, particularly for the second ($i=2$) and third ($i=3$) objectives?*
>
> **A6.** The experimental results only show that the regret of the first objective is smaller than that of the second, which is in turn smaller than that of the third. This empirical trend is consistent with the theoretical hierarchy across objective levels, but it does **not** allow us to confirm the exponential scaling implied by $\Lambda^i(\lambda)$.
>
> It is also worth noting that the theoretical bounds are **worst-case upper bounds**. The actual regret on a given instance may be substantially smaller, and numerical experiments do not necessarily correspond to the worst-case instances that realize the upper-bound scaling.
>
> ---
>
> Thank you again for your time and constructive feedback. We are happy to further discuss any aspect of our work.

---

> > ### Comment · Reviewer_W7Sg · 2025-11-24
> >
> > Thank you for your detailed response. Among all the points you addressed, I believe R1 and A1 are the most critical for evaluating this work. As the posterior sampling step is not actually required in either algorithm (MOMAB or MOSLB), Thompson Sampling cannot be considered a central algorithmic component of the paper. The core mechanism is an elimination-based procedure that balances exploration and exploitation, while the posterior distribution merely serves as a convenient tool for quantifying uncertainty. Given this, I am not able to revise my evaluation.

---

> > > ### Author Response · Authors · 2025-11-24
> > > **Many Thanks for the Follow-up**
> > >
> > > Thank you very much for your detailed follow-up and for taking the time to clarify your position. We sincerely appreciate your careful reading of our responses.
> > >
> > > Regarding R1, we agree that the issue was mainly a matter of terminology rather than substance. To avoid any further confusion, we have revised the presentation accordingly in the updated manuscript.
> > >
> > > For A1, our contribution lies in showing that when an elimination-based principle is implemented within a posterior-sampling (Thompson Sampling–style) framework, the resulting algorithm achieves regret guarantees that match the best-known results for this setting, while relying on a proof technique that is fundamentally different from existing TS analysis. In our view, a problem can benefit from having multiple algorithmic formulations and analysis, especially when they lead to complementary insights. The elimination-based interpretation provides an alternative lens that may be useful for extending posterior-sampling–driven approaches.
> > >
> > > We fully respect your evaluation and appreciate your constructive feedback, which has helped us improve the clarity and positioning of our work. Thank you again for your time and consideration.

---

### Official Review · Reviewer_WsPg · 2025-10-30

**Soundness:** 3
**Presentation:** 3
**Contribution:** 2
**Rating:** 2
**Confidence:** 3

**Summary:**

The paper considers a multi-objective version of the multiarmed bandit problem, where each arm is associated with a vector of reward distributions. In this setting, there are various possible notions of regret and the authors chose to define it with respect to a lexicographicall optimal arm. The authors give two algorithms, one for the case of standard  finite armed bandit setting with Gaussian rewards and the other for linear bandits with unknown reward distributions. There is some limited experimental evaluation.

**Strengths:**

- The paper is relatively well written
- It does a decent job of literature review.

**Weaknesses:**

Unfortunately i have several criticisms of the paper.

- The paper takes the notion of lexicographically optimal arms from [Huyuk and Tekin 2021] but their notion of regret is not the same.In particular, the present paper measures the regret for task $i$ of playing an arm with respect to the lexicographically optimal arm, which does not seems to make sense: more natural would be to compare  it against the reward of the optimal arm for that task. This is reflected also in the need to introduce the condition in (2) and (3), which seems contrived.
- The authors make no attempt to motivate the lexicographically optimal arm, nor their notion of regret. The original paper  [Huyuk and Tekin 2021] do give some motivating examples, but correspond to their notion of regret, not to this paper's.
- Algorithm 1 is not Thompson sampling at all!  It is closer to the Bayesian UCB in reference [Kaufmann et al 2021a]. The technical arguments seem to me quite similar, so there is limited novelty.
- The authors claim that methods based on Pareto dominance "assume equal importance across all objectives, ...” (lines 86–87). I don't think Pareto optimality makes any such assumption, but just recognize objectives are incomparable hence there is a tradeoff on the Pareto frontier.

**Questions:**

- Could you please clarify that your definition of regret is not he same as in [Huyuk and Tekin 2021] and if so, then justify why it is reasonable?
- Why do you call Algorith 1 Thompson sampling? isn't t closer to the Bayesian UCB from [Kaufmann et al 2021a]?

---

> ### Author Response · Authors · 2025-11-17
> **Response to Reviewer  WsPg: Part I**
>
> We sincerely appreciate the constructive feedback and have thoroughly evaluated the raised concerns. Our point-by-point responses to the weaknesses and questions follow below.
>
> ---
>
> **W1.** *The paper takes the notion of lexicographically optimal arms from [Huyuk and Tekin 2021] but their notion of regret is not the same. In particular, the present paper measures the regret for task $i$ of playing an arm with respect to the lexicographically optimal arm, which does not seems to make sense: more natural would be to compare it against the reward of the optimal arm for that task. This is reflected also in the need to introduce the condition in (2) and (3), which seems contrived.*
>
> **R1.** We thank the reviewer for suggesting to measure each objective’s regret against **the single-objective optimal arm for that objective**. Unfortunately, this suggestion is not feasible in the bandit setting if one expects sublinear regret for all objectives simultaneously. We give a simple lower-bound construction that illustrates the issue.
>
> Consider a 2-armed, 2-objective MOMAB with expected reward vectors $A=[1,0]$ and $B=[0,1]$. Under the reviewer’s proposed per-task regret,
>
> - the optimal arm for objective 1 is $A$ (reward 1) and the regret for objective 1 equals the number of times $B$ is played (each such play incurs regret $1-0=1$);
> - the optimal arm for objective 2 is $B$ (reward 1) and the regret for objective 2 equals the number of times $A$ is played.
>
> If the algorithm runs for $T$ rounds and plays $A$ exactly $n_A$ times and $B$ exactly $n_B$ times ($n_A+n_B=T$), then the two regrets are
>
> $$R^{1}(T) = n_B,\quad R^{2}(T) = n_A,$$
>
> hence
>
> $$R^{1}(T) + R^{2}(T) = n_A + n_B = T.$$
>
> Therefore at least one of $R^{1}(T)$ or $R^{2}(T)$ must be at least $T/2$, i.e. linear in $T$. This shows that no algorithm can achieve sublinear regret simultaneously for both objectives under the reviewer's per-task benchmark. In contrast, the lexicographic regret we use is well-defined and **allows sublinear regret**.
>
> ---
>
> **W2.** *The authors make no attempt to motivate the lexicographically optimal arm, nor their notion of regret. The original paper [Huyuk and Tekin 2021] do give some motivating examples, but correspond to their notion of regret, not to this paper's.*
>
> **R2.** We would like to clarify that our definition of the lexicographic optimal arm **is fully consistent with** that in [Hüyük and Tekin, 2021]: arms are compared sequentially according to the priority of objectives, and the arm with a higher expected reward in the most important differing objective is preferred.
>
> Regarding the **notion of regret**, our formulation follows the standard convention in the single-objective bandit literature, which measures the cumulative reward gap incurred by selecting suboptimal arms. Extending this definition to the lexicographic setting provides a natural and theoretically sound way to evaluate the performance of bandit algorithms under hierarchical objective preferences.
>
> ---
>
> **W3.** *Algorithm 1 is not Thompson sampling at all! It is closer to the Bayesian UCB in reference [Kaufmann et al 2012a]. The technical arguments seem to me quite similar, so there is limited novelty.*
>
> **R3.** We thank the reviewer for raising this important point. We acknowledge that the terminology in the initial submission was misleading, and we have corrected it in the revised version. Algorithm 1 is now referred to as **“Distribution-Known BayesUCB for Lexicographic Bandits (DK-BULB)”**, and all corresponding expressions have been updated accordingly.
>
> Regarding the novelty of the technical analysis: while it is true that many Bayesian-UCB-style analyses share a high-level structure, such as constructing confidence regions and studying their shrinkage rates. Our contribution lies in showing **how a Bayesian posterior-based strategy can be adapted to the lexicographic multi-objective setting**, which introduces several nontrivial challenges absent from single-objective bandits. In particular:
>
> - Lexicographic preferences require **hierarchical elimination across objectives**, and a key difficulty is ensuring that the optimal arm is not removed at any stage.
> - Our analysis leverages the observation that **all objectives share the same uncertainty width for a given arm**, which allows us to use the trade-off parameter $\lambda$ to guarantee consistent elimination across objectives—a property specific to multi-objective problems and non-obvious from prior single-objective work.
> - To the best of our knowledge, this is the **first regret analysis** showing that Bayesian posterior-based methods remain viable under lexicographic preference structures.

---

> > ### Comment · Reviewer_WsPg · 2025-11-23
> > **Response**
> >
> > Thanks for the simple example which is very clarifying and should be used in such papers to starkly illustrate the tradeoff. But it also shows that the more natural setup is one which allows the user to select their tradeoff between the two objectives: at the two extremes, the algorithm should achieve the optimal rate as in single task classical setting whereas if both tasks are equally important, then one cannot get sub-linear regret. I still don't see a good motivation for "lexicographcally optimal" arm.
> >
> > I'll revise my score accordingly.

---

> ### Author Response · Authors · 2025-11-17
> **Response to Reviewer WsPg: Part II**
>
> **W4.** *The authors claim that methods based on Pareto dominance "assume equal importance across all objectives, ...” (lines 86-87). I don't think Pareto optimality makes any such assumption, but just recognize objectives are incomparable hence there is a tradeoff on the Pareto frontier.*
>
> **R4.** We fully agree with the reviewer that Pareto optimality does **not** assume numerical equality of weights or imply that objectives are directly comparable. Our intent was to highlight the conceptual distinction between:
>
> - **lexicographic preferences**, which enforce a well-defined priority structure, and
> - **Pareto dominance**, which does not impose such a hierarchy and therefore treats objectives as having no pre-specified ordering.
>
> Given that multi-objective preferences are not uniquely defined, we acknowledge that different, equally valid interpretations exist. We have revised the wording to avoid potential ambiguity and to better reflect this distinction.
>
> ---
>
> **Q1.** *Could you please clarify that your definition of regret is not he same as in [Huyuk and Tekin 2021] and if so, then justify why it is reasonable?*
>
> **A1.** Yes, our regret definition **is different from** that in [Huyuk and Tekin, 2021]. Our formulation follows the **standard convention in the single-objective bandit literature**, where regret is defined as the cumulative reward gap incurred by choosing suboptimal arms.
>
> By extending this well-established single-objective notion to the lexicographic setting, using the lexicographically optimal arm as the benchmark, we obtain a natural and consistent performance measure. Therefore, as long as the classical regret definition is considered reasonable in the single-objective case, our lexicographic regret definition is also justified.
>
> ---
>
> **Q2.** *Why do you call Algorithm 1 Thompson sampling? isn't t closer to the Bayesian UCB from [Kaufmann et al 2012a]?*
>
> **R2.** We thank the reviewer for pointing this out. You are correct that Algorithm 1 aligns more closely with the Bayesian UCB approach of [Kaufmann et al., 2012a] rather than classical Thompson Sampling. We have updated the terminology accordingly in the revised version of the paper.
>
> ---
>
> Thank you again for your time and constructive feedback. We are happy to further discuss any aspect of our work.

---

> ### Author Response · Authors · 2025-11-23
> **Follow-up Example on the Motivation**
>
> Thank you very much for your thoughtful follow-up and for reconsidering your score. We sincerely appreciate your constructive comments.
>
> Regarding the motivation for the lexicographically optimal arm, our work focuses on scenarios in which the objectives have strict priority rather than a negotiable tradeoff. In many real-world decision-making processes such as safety-first optimization, users are not free to select an arbitrary tradeoff between objectives. Instead, certain objectives must be satisfied or optimized first, and only then can secondary objectives be considered. In such settings, lexicographic ordering provides a natural and principled way to capture the structure of the problem. The aim of our formulation is not to replace tradeoff-based multi-objective models, but to **characterize the theoretical limits and algorithmic principles for this important class of hierarchically structured preferences**.
>
> Furthermore, trade-off based approaches require the user to understand the shape of the Pareto front in order to choose a meaningful preference or weight vector. To illustrate this, consider two arms $A$ and $B$ evaluated on objectives $(f_1, f_2)$. Assume the true Pareto front is
>
> - Arm $A$: $f_1(A) = 0.90,\quad f_2(A) = 0.10.$
> - Arm $B$: $f_1(B) = 0.80,\quad f_2(B) = 0.60.$
>
> **Case I.** If the user selects a weight vector, say $w = (0.9, 0.1)$, without knowing the curvature of the Pareto front, then the weighted scores become
>
> - Weighted$(A) = 0.9 \times 0.90 + 0.1 \times 0.10 = 0.82.$
> - Weighted$(B) = 0.9 \times 0.80 + 0.1 \times 0.60 = 0.78.$
>
> **Arm$A$ is preferred.**
>
> **Case II.** But if the user chooses slightly different weights, such as $w = (0.8, 0.2)$, still intending that the first objective is more important, the ranking reverses:
>
> - Weighted$(A) = 0.8 \times 0.90 + 0.2 \times 0.10 = 0.74.$
> - Weighted$(B) = 0.8 \times 0.80 + 0.2 \times 0.60 = 0.76.$
>
> **Now arm $B$ is preferred.**
>
> This example shows that even small uncertainty about the weights can lead to entirely different decisions when using a trade-off strategy, because the correct choice depends sensitively on the local geometry of the Pareto front.
>
> ---
>
> Thank you once again for your helpful feedback and for your willingness to revise your score. If our additional clarification and example have helped to alleviate your concern about the motivation, we would sincerely appreciate your consideration of *a further positive adjustment*. We truly value your time and assessment.

---

### Official Review · Reviewer_pFDy · 2025-10-31

**Soundness:** 3
**Presentation:** 3
**Contribution:** 2
**Rating:** 4
**Confidence:** 3

**Summary:**

The submission investigates multi-objective bandit problems (MOMABs and MOSLBs) and introduces two Thompson Sampling (TS)-based algorithms: Distribution-Known Thompson Sampling for Lexicographic Bandits (DK-TSLB) and Distribution-Free Thompson Sampling for Lexicographic Bandits (DF-TSLB), tailored for MOMABs and MOSLBs, respectively. Theoretical analyses provide rigorous regret bounds, presented in Theorems 1 and 2, to substantiate the proposed methods. Furthermore, empirical simulations demonstrate results that are largely consistent with the theoretical findings.

**Strengths:**

- The submission explores the potential of TS based methods for multi-objective bandits.
- Concrete methods considering both distribution-known and distribution-unknown scenarios are proposed and analyzed rigorously.

**Weaknesses:**

- Two strict assumptions, the existence of a lexicographic optimal arm (LOA) and the validity of inequalities (2) and (3), limit the applicability of the proposed methods.
- Based on the intuition in (line 207), there may exist weaker and more realistic formulations to model the value improvement during solution transition.
- The key steps in the proofs (lines 760 and 911) directly invoke assumptions (2) and (3), making the analyses less technical.
- The results cannot be applied to the single-objective MAB case, as the regret bound becomes suboptimal; thus, one must rely on existing methods instead.
- The experiments do not include Gaussian reward functions to validate DK-TSLB.
- In the MOSLB experiments, finite arms are used instead of the theoretically assumed infinite arm setting, creating a discrepancy between theory and experiments.

**Questions:**

- It seems the word “trade-offs” in “capture the trade-offs between conflicting objectives, …” (line 200) implies the user has a preference among the objectives. Can this paper address user preference?
- Are assumptions (2) and the existence of LOA independent assumptions, or does one imply the other?
- In the proofs (Appendix B and C) of Theorems 1 and 2, at which specific steps is the LOA assumption utilized?

---

> ### Author Response · Authors · 2025-11-17
> **Response to Reviewer pFDy: Part I**
>
> We sincerely appreciate the constructive feedback and have thoroughly evaluated the raised concerns. Our point-by-point responses to the weaknesses and questions follow below.
>
> ---
>
> **W1.** *Two strict assumptions, the existence of a lexicographic optimal arm (LOA) and the validity of inequalities (2) and (3), limit the applicability of the proposed methods.*
>
> **R1.** Regarding the existence of the LOA, we note that the lexicographic order is a total order. Therefore, as long as the arm set is compact, a lexicographic optimal arm is guaranteed to exist. Specifically, in the MOMAB setting, the arm set is finite and hence compact, **ensuring the existence of an LOA**. In the MOSLB setting, if the **arm set is a bounded and closed** subset of the Euclidean space, compactness also holds, and consequently, the LOA exists. Thus, the existence assumption is not restrictive in practice.
>
> As for the assumptions (2) and (3), these assumptions are introduced to restrict the problem to a well-defined class of lexicographic bandit instances where rigorous theoretical analysis is tractable. Since no single algorithm can efficiently handle all possible multi-objective problems, assumptions (2) and (3) provide a principled way to capture the trade-offs between objectives and allow us to derive algorithms with provable regret guarantees.
>
> ---
>
> **W2.** *Based on the intuition in (line 207), there may exist weaker and more realistic formulations to model the value improvement during solution transition.*
>
> **R2.** At present, we are not aware of alternative weaker or more realistic formulations that can effectively replace assumptions (2) and (3) while still supporting a complete theoretical analysis. Nevertheless, we agree that relaxing these assumptions is an important research direction, and we are actively exploring ways to weaken them in future work.
>
> ---
>
> **W3.** *The key steps in the proofs (lines 760 and 911) directly invoke assumptions (2) and (3), making the analyses less technical.*
>
> **R3.** We appreciate the reviewer's careful reading of our proofs. We would like to emphasize two additional technical aspects:
>
> 1. The proofs of the corresponding lemmas (Lemmas 2 and 6) are established via mathematical induction, which is not a straightforward or intuitive approach in this context.
> 2. The reason assumptions (2) and (3) can be effectively utilized lies in a key observation: for a given arm, the estimates of all objectives are updated simultaneously, leading to identical confidence interval widths across objectives. This allows us to scale the confidence intervals using the trade-off parameter λ, ensuring that the optimal arm is not eliminated. This coupling among objectives is a distinctive feature of multi-objective problems and is nontrivial to exploit in the analysis.
>
> ---
>
> **W4.** *The results cannot be applied to the single-objective MAB case, as the regret bound becomes suboptimal; thus, one must rely on existing methods instead.*
>
> **R4.** We appreciate the reviewer’s comment. It is unclear to us in which parameter the reviewer considers our regret bound to be suboptimal. As shown in Table 1, for the MOMAB setting with known reward distributions, our algorithm achieves the same order in both $\Delta(a)$ and $T$ as the single-objective algorithm of Kaufmann et al. (2012b). Likewise, for the MOSLB setting with unknown distributions, our regret bound matches that of the single-objective algorithm Abeille and Lazaric (2017) in terms of $d$ and $T$. Therefore, our results are consistent with the optimal single-objective rates, and no degradation occurs when reducing to the single-objective case.
>
> ---
>
> **W5.** *The experiments do not include Gaussian reward functions to validate DK-TSLB.*
>
> **R5.** We believe the reviewer may have missed a relevant detail. As stated in Appendix A, our experiments already use **Gaussian reward distributions**: each stochastic reward $y_t^i$ is drawn from a normal distribution with mean $\mu^i(a)$ and variance $0.1$. Thus, DK-TSLB (DK-BULB)  is indeed evaluated under Gaussian rewards.

---

> ### Author Response · Authors · 2025-11-17
> **Response to Reviewer pFDy: Part II**
>
> **W6.** *In the MOSLB experiments, finite arms are used instead of the theoretically assumed infinite arm setting, creating a discrepancy between theory and experiments.*
>
> **R6.** We acknowledge the reviewer’s observation. We use a finite arm set in the MOSLB experiments for the following reasons:
>
> 1. *Fair comparison with MOMAB.* Using a finite arm set allows both MOMAB and MOSLB to be evaluated on the *same* problem instance, making the empirical comparison more meaningful and controlled.
> 2. *No essential difference for implementation.* Even if the arm set were infinite, in practice we would still construct a structured arm set (e.g., a ball or grid) so that the maximization steps in the algorithm admit exact solutions. This setup is conceptually equivalent to working with a finite discrete arm set.
> 3. *Common practice in the SLB literature.* Many existing stochastic linear bandit works conduct experiments on finite arm sets for the same practical reasons (e.g., [1–3]). Hence, our setup follows standard empirical practice in this domain.
>
> For these reasons, we believe the finite-arm design is appropriate for evaluating the MOSLB algorithms. We have clarified this motivation in the revised Appendix A.
>
> *Reference.*
> [1] Ruitu Xu, Yifei Min, and Tianhao Wang. Noise-adaptive thompson sampling for linear contextual
> bandits. NeurIPS, 2023.
> [2] Ziyi Huang, Henry Lam, and Haofeng Zhang. Bayesian Bandit Algorithms with Approximate Inference in Stochastic Linear Bandits. ArXiv 2025
> [3] Wonyoung Kim, Gi-Soo Kim, and Myunghee Cho Paik. Doubly Robust Thompson Sampling with Linear Payoffs. NeurIPS, 2021.
>
> ---
>
> **Q1.** *It seems the word “trade-offs” in “capture the trade-offs between conflicting objectives, …” (line 200) implies the user has a preference among the objectives. Can this paper address user preference?*
>
> **A1.** Yes, our framework can incorporate user preferences, but it does so through a strict lexicographic ordering rather than through weight-based preference modeling. Under this structure, the $i$-th objective is infinitely more important than the $(i+1)$-th one. In this sense, lexicographic preferences represent a special form of user preference with absolute priority among objectives.
>
> The parameter $\lambda$ in assumptions (2) and (3) **does not encode** user preferences directly, **but instead** quantifies the difficulty of identifying the optimal arm. For example, consider two expected reward vectors $[1, 0.9]$ and $[0.9, 5]$, whose $\lambda = 49$.  Under uncertainty, the large trade-off makes it tend to misidentify $[0.9, 5]$ as optimal. In contrast, when comparing $[1, 0.9]$ and $[0.9, 0.9]$, where $\lambda = 0$, the true optimal arm $[1, 0.9]$ can be identified much more easily. Thus, $\lambda$ reflects the strength of trade-offs between objectives and characterizes how these trade-offs affect the identification of the lexicographic optimal arm, rather than modeling user preference explicitly.
>
> ---
>
> **Q2.** *Are assumptions (2) and the existence of LOA independent assumptions, or does one imply the other?*
>
> **A2.** They are not independent. The validity of assumption (2) relies on the existence of a lexicographic optimal arm (LOA). In other words, once assumption (2) holds, the LOA is guaranteed to exist.
>
> ---
>
> **Q3.** *In the proofs (Appendix B and C) of Theorems 1 and 2, at which specific steps is the LOA assumption utilized?*
>
> **A3.** The existence of the LOA is a fundamental prerequisite for our analysis. It is not used in a specific step of the proofs but serves as the foundation of the entire regret analysis, since the regret is defined with respect to the lexicographic optimal arm. Without the existence of the LOA, the notion of regret would be meaningless.
>
> ---
>
> Thank you again for your time and constructive feedback. We are happy to further discuss any aspect of our work.

---

> > ### Comment · Reviewer_pFDy · 2025-11-21
> > **Thank you for the response**
> >
> > Thank you for the reply, especially for W1 and W6. Reply 2. in R3 is indeed nontrivial, but the total contributions are not substantial. Thus, I would like to keep my score for now. Thank you.

---

> > > ### Author Response · Authors · 2025-11-21
> > > **Follow-up Response to Reviewer**
> > >
> > > Thank you for your follow-up assessment. We appreciate the time and effort you have devoted to reviewing our rebuttal and manuscript. We respect your decision to maintain the current score.
> > >
> > > If there is any remaining concern, we would be glad to elaborate. Otherwise, if our rebuttal has meaningfully reduced the earlier concerns, we would sincerely appreciate your consideration of adjusting the score accordingly.
> > >
> > > Thank you again for your thoughtful review.

---

### Note · Authors · 2026-01-07

**Comment:**

We would like to express our sincere gratitude to the reviewers for their insightful comments and constructive suggestions. We also appreciate the efforts of the Area Chairs and the organizing committee in coordinating the review process.

After careful consideration of the feedback, we have decided to withdraw the paper to further refine our work based on the reviewers' recommendations.

**Withdrawal Confirmation:**

I have read and agree with the venue's withdrawal policy on behalf of myself and my co-authors.